# Breaking Reversibility Accelerates Langevin Dynamics for Non-Convex Optimization

**Xuefeng Gao**[*]
Department of Systems Engineering and Engineering Management
The Chinese University of Hong Kong
Shatin, N.T. Hong Kong
`xfgao@se.cuhk.edu.hk`

**Mert Gürbüzbalaban**[*]
Department of Management Science and Information Systems
Rutgers Business School
Piscataway, NJ-08854, United States of America
`mg1366@rutgers.edu`

**Lingjiong Zhu**[*]
Department of Mathematics
Florida State University
Tallahassee, FL-32306, United States of America
`zhu@math.fsu.edu`

## Abstract

Langevin dynamics (LD) has been proven to be a powerful technique for optimizing a non-convex objective as an efficient algorithm to find local minima while eventually visiting a global minimum on longer time-scales. LD is based on the first-order Langevin diffusion which is reversible in time. We study two variants that are based on non-reversible Langevin diffusions: the underdamped Langevin dynamics (ULD) and the Langevin dynamics with a non-symmetric drift (NLD). Adopting the techniques of Tzen et al. (2018) for LD to non-reversible diffusions, we show that for a given local minimum that is within an arbitrary distance from the initialization, with high probability, either the ULD trajectory ends up somewhere outside a small neighborhood of this local minimum within a recurrence time which depends on the smallest eigenvalue of the Hessian at the local minimum or they enter this neighborhood by the recurrence time and stay there for a potentially exponentially long escape time. The ULD algorithm improves upon the recurrence time obtained for LD in Tzen et al. (2018) with respect to the dependency on the smallest eigenvalue of the Hessian at the local minimum. Similar results and improvements are obtained for the NLD algorithm. We also show that non-reversible variants can exit the basin of attraction of a local minimum faster in discrete time when the objective has two local minima separated by a saddle point and quantify the amount of improvement. Our analysis suggests that non-reversible Langevin algorithms are more efficient to locate a local minimum as well as exploring the state space.

---

[*]The authors are in alphabetical order.

Consider the stochastic optimization problem:

$$\min_{x \in \mathbb{R}^d} \overline{F}(x) := \mathbb{E}_{Z \sim P}[f(x, Z)] = \int_{\mathcal{Z}} f(x, z) P(dz),$$

where $f : \mathbb{R}^d \times \mathcal{Z} \to \mathbb{R}$ is a real-valued, smooth, possibly non-convex objective function with two inputs, the decision vector $x \in \mathbb{R}^d$ and a random vector $Z$ with probability distribution $P$ defined on a set $\mathcal{Z}$. A standard approach for solving stochastic optimization problems is to approximate the expectation as an average over independent observations $z = (z_1, z_2, \ldots, z_n) \in \mathcal{Z}^n$ and to solve:

$$\min_{x \in \mathbb{R}^d} F(x) := \frac{1}{n} \sum_{i=1}^{n} f(x, z_i). \tag{0.1}$$

Such problems with finite-sum structure arise in many applications, e.g. data analysis and machine learning ([GBCB16]). In this work, our primary focus will be non-convex objectives.

First-order methods such as gradient descent and stochastic gradient descent and their variants with momentum have been popular for solving such optimization problems (see e.g. [Ber15, Bub14]). These first-order methods admit some theoretical guarantees to locate a local minimizer, however their convergence depends strongly on the initialization and they do not have guarantees to visit a global minimum. The Langevin Dynamics (LD) is a variant of gradient descent where a properly scaled Gaussian noise is added to the gradients:

$$X_{k+1} = X_k - \eta \nabla F(X_k) + \sqrt{2\eta\beta^{-1}} \xi_k,$$

where $\eta > 0$ is the stepsize, $\xi_k$ is a $d$-dimensional isotropic Gaussian noise with distribution $\mathcal{N}(0, I)$ where for every $k$, the noise $\xi_k$ is independent of the (filtration) past up to time $k$ and $\beta > 0$ is called the *inverse temperature* parameter. With proper choice of parameters and under mild assumptions, LD algorithm converges to a stationary distribution that concentrates around a global minimum (see e.g. [BM99, GM91]) from an arbitrary initial point. Therefore, LD algorithm has a milder dependency on the initialization, visiting a global minimum eventually. The analysis of the convergence behavior of LD is often based on viewing LD as a discretization of the associated stochastic differential equation (SDE), known as the *overdamped Langevin* diffusion or the *first-order Langevin* diffusion,

$$dX(t) = -\nabla F(X(t))dt + \sqrt{2\beta^{-1}} dB_t, \tag{0.2}$$

where $B_t$ is a $d$-dimensional standard Brownian motion (see e.g. [GM91]). Under some mild assumptions on $F$, this SDE admits the following unique stationary distribution:

$$\pi(dx) = \Gamma^{-1} e^{-\beta F(x)} dx, \tag{0.3}$$

where $\Gamma > 0$ is a normalizing constant. Note that overdamped Langevin diffusion is *reversible* in the sense that if $X(0)$ is distributed according to the stationary measure $\pi$, then $(X_t)_{0 \le t \le T}$ and $(X_{T-t})_{0 \le t \le T}$ have the same law. It is known that the reversible Langevin algorithm converges to a local minimum in time polynomial with respect to parameters $\beta$ and $d$, the intuition being that the expectation of the iterates follows the gradient descent dynamics which converges to a local minimum (see e.g. [ZLC17, FGQ97]). It is also known that once Langevin algorithms arrive to a neighborhood of a local optimum, they can spend exponentially many iterations in dimension to escape from the basin of attraction of this local minimum. This behavior is known as "metastability" and has been studied well (see e.g. [BGK05, BGK04, Ber13]).

Recently, [TLR18] provided a finer characterization of this metastability phenomenon. They showed that for a given local optimum $x_*$, with high probability and arbitrary initialization, either LD iterates arrive at a point outside an $\varepsilon$-neighborhood of this local minimum within a recurrence time $\mathcal{T}_{\text{rec}} = \mathcal{O}\left(\frac{1}{m} \log(\frac{1}{\varepsilon})\right)$, where $m$ is smallest eigenvalue of the Hessian $\nabla^2 F(x_*)$ at the local minimum or they enter this $\varepsilon$-neighborhood by the recurrence time and stay there until a potentially exponentially long escape time $\mathcal{T}_{\text{esc}}$. The escape time $\mathcal{T}_{\text{esc}}$ measures how quickly the LD algorithm can get away from a given neighborhood around a local minimum, therefore it can be viewed as a measure of the effectiveness of LD for the search of a global minimizer, whereas the recurrence time $\mathcal{T}_{\text{rec}}$ can be viewed as the order of the time-scale for which search for local minima in the basin of attraction of that minimum happens.

One popular non-reversible variant of overdamped Langevin that can improve its performance in practice for a variety of applications (Section 4 in [LNP13]) is based on the *underdamped Langevin*

diffusion, also known as the *second-order Langevin* diffusion ([Kra40]),

$$dV(t) = -\gamma V(t)dt - \nabla F(X(t))dt + \sqrt{2\gamma\beta^{-1}}dB_t, \tag{0.4}$$
$$dX(t) = V(t)dt, \tag{0.5}$$

where $X(t), V(t) \in \mathbb{R}^d$, and $\gamma > 0$ is known as the *friction coefficient*. It is known that under mild assumption on $F$, the Markov process $(X, V)$ is ergodic and have a unique stationary distribution

$$\pi(dx, dv) = \Gamma_U^{-1} e^{-\beta(\frac{1}{2}\|v\|^2 + F(x))} dx dv, \tag{0.6}$$

where $\Gamma_U > 0$ is a normalizing constant. Hence, the marginal distribution in $X$ of the Gibbs distribution $\pi(dx, dv)$ is the same as the invariant distribution (0.3) of the overdamped Langevin dynamics (0.2). We refer the readers to [BCG08, CCBJ18, CCA$^+$18, DRD20, EGZ19, MCC$^+$19, MSH02, Wu01, Vil09] for more on underdamped Langevin diffusions. The Euler discretization of the underdamped Langevin diffusion is known as the inertial Langevin dynamics or the Hamiltonian Langevin Monte Carlo algorithm [DKPR87, Nea10]. [CCBJ18] introduced a more accurate discretization of underdamped Langevin diffusion, where for any $k \in \mathbb{N}$ and any $k\eta < t \leq (k+1)\eta$,

$$d\tilde{V}(t) = -\gamma\tilde{V}(t)dt - \nabla F(\tilde{X}(k\eta))dt + \sqrt{2\gamma\beta^{-1}}dW_t, \qquad d\tilde{X}(t) = \tilde{V}(t)dt.$$

Note that when $t$ is between $k\eta$ and $(k+1)\eta$, the above diffusion process is an Ornstein-Uhlenbeck process, which is a Gaussian process with explicit mean and covariance. [CCBJ18] showed that $(V_k, X_k)$ have the same distribution as $(\tilde{V}(k\eta), \tilde{X}(k\eta))$, where the discrete iterates $(V_k, X_k)$, called the *underdamped Langevin dynamics* (ULD)[2] are generated as follows:

$$V_{k+1} = \psi_0(\eta)V_k - \psi_1(\eta)\nabla F(X_k) + \sqrt{2\gamma\beta^{-1}}\xi_{k+1}, \tag{0.7}$$
$$X_{k+1} = X_k + \psi_1(\eta)V_k - \psi_2(\eta)\nabla F(X_k) + \sqrt{2\gamma\beta^{-1}}\xi'_{k+1}, \tag{0.8}$$

where $(\xi_{k+1}, \xi'_{k+1})$ is a $2d$-dimensional centered Gaussian vector so that $(\xi_j, \xi'_j)$ are i.i.d. and independent of the initial condition, and for any fixed $j$, the random vectors $((\xi_j)_1, (\xi'_j)_1), ((\xi_j)_2, (\xi'_j)_2),$ $\ldots, ((\xi_j)_d, (\xi'_j)_d)$ are i.i.d. with the covariance matrix: $C(\eta) = \int_0^\eta [\psi_0(t), \psi_1(t)]^T [\psi_0(t), \psi_1(t)]dt$, where $\psi_0(t) = e^{-\gamma t}$ and $\psi_{k+1}(t) = \int_0^t \psi_k(s)ds$ for every $k \geq 0$. Recent work [GGZ18] showed that ULD admits better non-asymptotic performance guarantees compared to known guarantees for LD in the context of non-convex optimization when the objective satisfies a dissipativity condition. Recent work also showed that ULD or alternative discretizations of the underdamped diffusion can sample from the Gibbs distribution more efficiently than LD when $F$ is globally strongly convex (see e.g. [CCBJ18, DRD20, MS17]) or strongly convex outside a compact set (see e.g. [CCA$^+$18]).

The second non-reversible variant of overdamped Langevin involves adding a drift term:

$$dX(t) = -A_J(\nabla F(X(t)))dt + \sqrt{2\beta^{-1}}dB_t, \qquad A_J := I + J, \tag{0.9}$$

where $J \neq 0$ is a $d \times d$ anti-symmetric matrix, i.e. $J^T = -J$ and $I$ is the $d \times d$ identity matrix, and $B_t$ is a standard $d$-dimensional Brownian motion. It can be shown that such a drift preserves the stationary distribution (0.3) (Gibbs distribution) of the overdamped Langevin dynamics, and it can lead to a faster convergence to the stationary distribution than the reversible Langevin diffusion (the case $J = 0$), see e.g. [HHMS93, HHMS05, LNP13, Pav14, GM16] for details. Algorithms based on (0.9) have been applied to sampling, see e.g. [FSS20, RBS16, RBS15, DLP16, DPZ17], and non-convex optimization, see e.g. [HWG$^+$20]. The Euler discretization of (0.9) leads to

$$X_{k+1} = X_k - \eta A_J(\nabla F(X_k)) + \sqrt{2\eta\beta^{-1}}\xi_k, \tag{0.10}$$

which we refer to as the *non-reversible Langevin dynamics* (NLD).

**Contributions.** In this paper, we investigate the metastability behavior of non-reversible Langevin algorithms for non-convex objectives. We extend the results of [TLR18] to non-reversible Langevin dynamics and show that for a given local minimum that is within an arbitrary distance $r$ from the initialization, with high probability, either ULD trajectory ends up somewhere outside an $\varepsilon$-neighborhood of this local minimum within a recurrence time $\mathcal{T}_{\text{rec}}^U = \mathcal{O}\left(\frac{|\log(m)|}{\sqrt{m}}\log(r/\varepsilon)\right)$ or they enter this neighborhood by the recurrence time and stay there for a potentially exponentially long

escape time. The analogous result shown in [TLR18] for reversible LD requires a recurrence time of $\mathcal{T}_{\text{rec}} = \mathcal{O}\left(\frac{1}{m}\log(r/\varepsilon)\right)$. This shows that underdamped dynamics requires a smaller recurrence time by a square root factor in $m$ (ignoring a $\log(m)$ factor). The difference is significant as the smallest eigenvalue $m$ of the Hessian matrix at a local optimum can be very small in a number of applications, including deep learning (see e.g. [CCS$^{+}$17, SBL16]). Since the recurrence time can be viewed as a measure of the efficiency of the search of a local minimum [TLR18], our results suggest that ULD operates on a faster time-scale to locate a local minimum. Similar results are obtained for NLD. In order to obtain the results, we first give a refined characterization of the dynamics around a local minimum by linearizing the gradients. The analysis here is more complicated compared to the LD case in [TLR18] due to non-reversibility, and requires us to develop new estimates, e.g. Lemma 2, where the eigenvalue and the norm estimates require a significant amount of work because the forward iterations correspond to non-symmetric matrices $H_\gamma$ (defined in (1.3)) and achieving the acceleration behavior requires careful estimates. The analysis here also requires us to establish novel uniform $L^2$ bounds for NLD in both continuous and discrete times.

In addition, we consider the mean exit time from the basin of attraction of a local minimum for non-reversible algorithms. We focus on the double-well example which has been the recent focus of the literature [BR16, LMS19] as it is the simplest non-convex function that gives intuition about the more general case and for which mean exit time has been studied in continuous time. Our analysis shows that non-reversible dynamics can exit the basin of attraction of a local minimum faster under some conditions and characterizes the improvement for both ULD and NLD compared to LD when the parameters of these algorithms are chosen appropriately. These results support the numerical evidence that non-reversible algorithms can explore the state space more efficiently [CDC15, CFG14, GM16] and bridges a gap between the theory and practice of Langevin algorithms.

**Other related work.** Langevin dynamics has been studied under simulated annealing algorithms in the optimization, physics and statistics literature and its asymptotic convergence guarantees are well known (see e.g. [Gid85, Haj85, GM91, KGV83, BT93, BLNR15, BM99]). However, finite-time performance guarantees for LD have not been studied until more recently (see e.g. [Dal17, DM17]). Non-asymptotic performance guarantees for stochastic gradient versions have also been studied. See also e.g. [RRT17, ZLC17, CDT20] for related results. [XCZG18] shows that it suffices to have $\mathcal{O}(nd/(\lambda\varepsilon))$ gradient evaluations or $\mathcal{O}(d^7/(\lambda^5\varepsilon^5))$ stochastic gradient evaluations to compute an almost minimizer where $\lambda$ is a spectral gap parameter that is exponentially small in the dimension $d$ and $\varepsilon$ is the target accuracy. These results improve upon the existing results from the seminal work of [RRT17]. [EMS18] also considered Euler discretization of general dissipative diffusions in the non-convex setting, proved a $1/\varepsilon^2$ convergence rate, showing that different diffusions are suitable for minimizing different convex/non-convex objective functions $f$. Their expected suboptimality bound also generalizes the results in [RRT17]. See also [NŞR19] for non-asymptotic guarantees for non-convex optimization using Lévy-driven Langevin dynamics proposed in [Şim17].

**Notations.** Throughout the paper, for any $x, y \in \mathbb{R}$, we use the notation $x \wedge y$ to denote $\min\{x, y\}$ and $x \vee y$ to denote $\max\{x, y\}$. For any $n \times n$ matrix $A$, we use $\lambda_i(A)$, $1 \leq i \leq n$, to denote the $n$ eigenvalues of $A$. We also assume that $H$ is the Hessian matrix $\nabla^2 F$ evaluated at the local minimum $x_*$, and is positive definite. The norm $\|\cdot\|$ denotes the 2-norm of a vector and the (spectral norm) 2-norm of a matrix.

# 1 Main results

In this section, we will study the recurrence time $\mathcal{T}_{\text{rec}}^U$ of underdamped Langevin dynamics (ULD), and the corresponding time-scale $\mathcal{T}_{\text{rec}}^J$ for non-reversible Langevin dynamics (NLD). We will show that recurrence time of underdamped and non-reversible Langevin algorithms will improve upon that of reversible Langevin algorithms in terms of its dependency to the smallest eigenvalue of the Hessian at a local minimum. For non-convex optimizations, our results suggest that for ULD and NLD, searching for a local minimum happens on a faster time scale compared to the reversible LD. For the rest of the paper, we impose the following assumptions.

**Assumption 1.** *We impose the following assumptions.*

&emsp;(*i*) *The functions $f(\cdot, z)$ are twice continuously differentiable, non-negative valued, and $|f(0, z)| \leq A$, $\|\nabla f(0, z)\| \leq B$, and $\left\|\nabla^2 f(0, z)\right\| \leq C$, uniformly in $z \in \mathcal{Z}$ for some $A, B, C > 0$.*

(ii) $f(\cdot, z)$ *have Lipschitz-continuous gradients and Hessians, uniformly in $z \in \mathcal{Z}$, there exist constants $L, M > 0$ so that for all $x, y \in \mathbb{R}^d$,*

$$\|\nabla f(x, z) - \nabla f(y, z)\| \leq M\|x-y\|, \qquad \left\|\nabla^2 f(x, z) - \nabla^2 f(y, z)\right\| \leq L\|x-y\|. \quad (1.1)$$

(iii) *The empirical risk $F(\cdot)$ is $(m, b)$-dissipative: $\langle x, \nabla F(x)\rangle \geq m\|x\|^2 - b$.*

(iv) *The initialization satisfies $\|X_0\| \leq R := \sqrt{b/m}$.*

For the Hessian $H$ at a fixed local minimum of $F$ defined in (0.1), it is a $d \times d$ symmetric positive definite matrix with eigenvalues $\{\lambda_i\}_{i=1}^d$ in increasing order, i.e.[3]

$$m = \lambda_1 \leq \lambda_2 \leq \cdots \leq \lambda_d = M. \quad (1.2)$$

## 1.1 Underdamped Langevin dynamics

Recall the underdamped Langevin (0.7)-(0.8). Define

$$H_\gamma := \left[ \begin{array}{cc} \gamma I & H \\ -I & 0 \end{array} \right], \quad (1.3)$$

where we recall that $H$ is the Hessian matrix $\nabla^2 F$ evaluated at the local minimum $x_*$. In the first lemma, we provide an estimate on $\|e^{-tH_\gamma}\|$. This is the key result that allows the underdamped dynamics to achieve faster rate compared with overdamped dynamics.

**Lemma 2** (Estimate on $\|e^{-tH_\gamma}\|$). *(i) If $\gamma \in (0, 2\sqrt{m})$, then $\left\|e^{-tH_\gamma}\right\| \leq C_{\hat\varepsilon} e^{-\sqrt{m}(1-\hat\varepsilon)t}$, where $C_{\hat\varepsilon} := \frac{1+M}{\sqrt{m(1-(1-\hat\varepsilon)^2)}}$, and $\hat\varepsilon := 1 - \frac{\gamma}{2\sqrt{m}} \in (0, 1)$, where $m, M$ are defined in (1.2). (ii) If $\gamma = 2\sqrt{m}$, then we have $\left\|e^{-tH_\gamma}\right\| \leq \sqrt{C_H + 2 + (m+1)^2 t^2} \cdot e^{-\sqrt{m}t}$, where $C_H := \max_{i:\lambda_i > m} \frac{(1+\lambda_i)^2}{\lambda_i - m}$.*

We investigate the behavior around local minima for the underdamped Langevin dynamics (0.7)-(0.8) by studying recurrence and escape times with the choice of the friction coefficient $\gamma = 2\sqrt{m}$ which is optimal for $\|e^{-tH_\gamma}\|$. We first recall the Lambert W function $W(x)$ which is defined via the solution of the algebraic equation $W(x)e^{W(x)} = x$. When $-e^{-1} \leq x < 0$, $W(x)$ has two branches, the upper branch $W_0(x)$ and the lower branch $W_{-1}(x)$, see e.g. [CGH+96].

**Theorem 3.** *Fix $\gamma = 2\sqrt{m}$, $\delta \in (0, 1)$ and $r > 0$. For a given $\varepsilon$ satisfying $0 < \varepsilon < \bar{\varepsilon}^U = \min\{\mathcal{O}(r), \mathcal{O}(m)\}$, we define the recurrence time*

$$\mathcal{T}_{rec}^U = -\frac{1}{\sqrt{m}} W_{-1}\left( \frac{-\varepsilon^2 \sqrt{m}}{8r^2 \sqrt{C_H + 2 + (m+1)^2}} \right) = \mathcal{O}\left( \frac{|\log(m)|}{\sqrt{m}} \log\left(\frac{r}{\varepsilon}\right) \right),$$

*and the escape time $\mathcal{T}_{esc}^U := \mathcal{T}_{rec}^U + \mathcal{T}$, for any arbitrary $\mathcal{T} > 0$. Consider an arbitrary initial point $x$ for the underdamped Langevin dynamics and a local minimum $x_*$ at a distance at most $r$. Assume that the stepsize $\eta$ satisfies*

$$\eta \leq \bar{\eta}^U = \min\left\{ \mathcal{O}(\varepsilon), \mathcal{O}\left( \frac{m^2 \beta \delta \varepsilon^2}{(md + \beta)\mathcal{T}_{rec}^U} \right), \mathcal{O}\left( \frac{m^{5/4}\delta}{(d + \beta)^{1/2}(\mathcal{T}_{esc}^U)^{1/2}} \right), \mathcal{O}(m^{5/2}) \right\},$$

*and $\beta$ satisfies*

$$\beta \geq \underline{\beta}^U = \max\left\{ \Omega\left( \frac{d + \log((\mathcal{T}+1)/\delta)}{m\varepsilon^2} \right), \Omega\left( \frac{d\eta m^{1/2} \log(\delta^{-1}\mathcal{T}_{rec}^U/\eta)}{\varepsilon^2} \right) \right\},$$

*for any realization of training data $z$, with probability at least $1 - \delta$ with respect to the Gaussian noise, at least one of the following events will occur: (1) $\|X_k - x_*\| \geq \frac{1}{2}\left( \varepsilon + re^{-\sqrt{m}k\eta} \right)$ for some $k \leq \eta^{-1}\mathcal{T}_{rec}^U$; (2) $\|X_k - x_*\| \leq \varepsilon + re^{-\sqrt{m}k\eta}$ for every $\eta^{-1}\mathcal{T}_{rec}^U \leq k \leq \eta^{-1}\mathcal{T}_{esc}^U$.*

The expressions of technical constants in the statement of Theorem 3, including $\bar{\varepsilon}^U, \bar{\eta}^U$ and $\underline{\beta}^U$, can be found in the proof of Theorem 3 in the Supplementary File.

In many applications, the eigenvalues of the Hessian at local extrema often concentrate around zero and the magnitude of the smallest eigenvalue $m$ of the Hessian can be very small (see e.g. [CCS$^+$17, SBL16]). In [TLR18], the overdamped Langevin algorithm is analyzed and the recurrence time $\mathcal{T}_{\text{rec}} = \mathcal{O}\left(\frac{1}{m}\log(\frac{r}{\varepsilon})\right)$, while our recurrence time $\mathcal{T}_{\text{rec}}^U = \mathcal{O}\left(\frac{|\log(m)|}{\sqrt{m}}\log(\frac{r}{\varepsilon})\right)$ for the underdamped Langevin algorithm, which has a square root factor improvement. Since the recurrence time can be viewed as a measure of the efficiency of the search of a local minimum, our result suggests that ULD require smaller recurrence time, so they operate on a faster time scale to locate a local minimum.

## 1.2 Non-reversible Langevin dynamics

We investigate the behavior around local minima for the non-reversible Langevin dynamics (0.10) by studying recurrence and escape times. One can expect that the convergence behavior of the non-reversible Langevin diffusion is controlled by the decay of $\left\|e^{-tA_J H}\right\|$ in time $t$, which is related to the real part of the eigenvalues $\lambda_i^J := \text{Re}\left(\lambda_i(A_J H)\right)$ indexed with increasing order and their multiplicity. It has been shown that for any anti-symmetric matrix $J$, we have $m = \lambda_1 \leq \lambda_1^J \leq \lambda_d^J \leq \lambda_d = M$, and $m = \lambda_1 = \lambda_1^J$ if a very special condition holds. See Theorem 3.3. in [HHMS93] for details. This suggests generically non-reversible Langevin leads to a faster exponential decay compared to reversible Langevin, i.e $\lambda_1^J > \lambda_1$. In addition, we have the following estimates: there exists a positive constant $C_J$ that depends on $J$ such that

$$\left\|e^{-tA_J H}\right\| \leq C_J(1 + t^{n_1-1})e^{-t\lambda_1^J}, \tag{1.4}$$

where $n_1$ is the maximal size of a Jordan block of $A_J H$ corresponding to the eigenvalue $\lambda_1^J$. It follows that for any $\tilde{\varepsilon} > 0$, there exist some constant $C_J(\tilde{\varepsilon})$ that depends on $\tilde{\varepsilon}$ and $J$ such that for every $t \geq 0$,

$$\left\|e^{-tA_J H}\right\| \leq C_J(\tilde{\varepsilon})e^{-tm_J(\tilde{\varepsilon})}, \qquad m_J(\tilde{\varepsilon}) := \lambda_1^J - \tilde{\varepsilon}. \tag{1.5}$$

Now we state the main result on the metastability of non-reversible Langevin dynamics (0.10).

**Theorem 4.** *Fix $\delta \in (0,1)$ and $r > 0$. For a given $\varepsilon$ satisfying $0 < \varepsilon < \bar{\varepsilon}^J = \min\left\{\mathcal{O}\left(\frac{m_J(\tilde{\varepsilon})}{C_J(\tilde{\varepsilon})}\right), \mathcal{O}(rC_J(\tilde{\varepsilon}))\right\}$, we define the recurrence time*

$$\mathcal{T}_{rec}^J := \frac{2}{m_J(\tilde{\varepsilon})}\log\left(\frac{8r}{C_J(\tilde{\varepsilon})\varepsilon}\right) = \mathcal{O}\left(\frac{1}{m_J(\tilde{\varepsilon})}\log\left(\frac{r}{C_J(\tilde{\varepsilon})\varepsilon}\right)\right),$$

*and the escape time $\mathcal{T}_{esc}^J := \mathcal{T}_{rec}^J + \mathcal{T}$ for any arbitrary $\mathcal{T} > 0$. For any initial point $x$ and a local minimum $x_*$ at a distance at most $r$. Assume the stepsize*

$$\eta \leq \bar{\eta}^J = \min\left\{\mathcal{O}\left(\varepsilon\right), \mathcal{O}\left(\frac{\delta\varepsilon^2 m^3}{(m + \beta^{-1}d)\mathcal{T}_{rec}^J}\right), \mathcal{O}\left(\frac{\delta^2 m^3}{(d + m\beta + dm^3)\mathcal{T}_{esc}^J}\right)\right\},$$

*and $\beta$ satisfies*

$$\beta \geq \underline{\beta}^J = \max\left\{\Omega\left(\frac{C_J(\tilde{\varepsilon})^2}{m_J(\tilde{\varepsilon})\varepsilon^2}\left(d + \log\left(\frac{\mathcal{T}+1}{\delta}\right)\right)\right), \Omega\left(\frac{d\eta\log(\delta^{-1}\mathcal{T}_{rec}^J/\eta)}{\varepsilon^2}\right)\right\},$$

*for any realization of training data $z$, with probability at least $1 - \delta$ with respect to the Gaussian noise, at least one of the following events will occur: (1) $\|X_k - x_*\| \geq \frac{1}{2}\left(\varepsilon + re^{-m_J(\tilde{\varepsilon})k\eta}\right)$ for some $k \leq \eta^{-1}\mathcal{T}_{rec}^J$; (2) $\|X_k - x_*\| \leq \varepsilon + re^{-m_J(\tilde{\varepsilon})k\eta}$ for every $\eta^{-1}\mathcal{T}_{rec}^J \leq k \leq \eta^{-1}\mathcal{T}_{esc}^J$.*

The expressions of technical constants in the statement of Theorem 4, including $\bar{\varepsilon}^J, \bar{\eta}^J$ and $\underline{\beta}^J$, can be found in the proof of Theorem 4 in the Supplementary File.

In [TLR18], the overdamped Langevin algorithm is used and the recurrence time $\mathcal{T}_{\text{rec}} = \mathcal{O}\left(\frac{1}{m}\log(\frac{r}{\varepsilon})\right)$, while our recurrence time $\mathcal{T}_{\text{rec}}^J = \mathcal{O}\left(\frac{1}{m_J(\tilde{\varepsilon})}\log(\frac{r}{C_J(\tilde{\varepsilon})\varepsilon})\right)$ for the non-reversible Langevin algorithm, and $\mathcal{T}_{\text{rec}}^J = \mathcal{O}\left(\frac{1}{m_J(\tilde{\varepsilon})}\log(\frac{r}{\varepsilon})\right)$ when $C_J(\tilde{\varepsilon}) = \mathcal{O}(1)$, which has the improvement over the overdamped Langevin algorithm since $m_J(\tilde{\varepsilon}) > m$ in general.

One could also ask what is the choice of the matrix $J$ in NLD. A natural idea is to maximize the exponent $\lambda_1^J$ that appears in Equation (1.4), i.e., let $J_{opt} := \arg\max_{J=-J^T} \lambda_1^J$. A formula for $J_{opt}$ and an algorithm to compute it is known (see Fig. 1 in [LNP13]), however this is not practical to compute for optimization purposes as it requires the knowledge of the eigenvectors and eigenvalues of the matrix $H$ which is generally unknown in practice. Nevertheless, $J_{opt}$ gives information about the extent of acceleration that can be obtained. It is known that $\lambda_1^{J_{opt}} = \frac{\text{Tr}(H)}{d}$, as well as a characterization of the constants $C_{J_{opt}}$ and $n_1$ arising in Equation (1.4) when $J = J_{opt}$ (see Equation (46) in [LNP13]). We see that $md \leq \text{Tr}(H) \leq M(d-1) + m$ as the smallest and the largest eigenvalue of $H$ is $m$ and $M$. Therefore, we have

$$1 \leq \frac{\lambda_1^{J_{opt}}}{\lambda_1} \leq \frac{M(d-1) + m}{md}.$$

The acceleration is not possible (the ratio above is 1) if and only if all the eigenvalues of $H$ are the same and are equal to $m$; i.e. when $M = m$ and $\text{Tr}(H) = md$. Otherwise, $J_{opt}$ can accelerate by a factor of $\frac{M(d-1)+m}{md}$ which is on the order of the condition number $\kappa := M/m$ up to a constant $\frac{d-1}{d}$ which is close to one for $d$ large. In practice, one can also use some easily constructed choices of $J$ as suggested in the literature (see e.g. [HHMS93]), and run the NLD algorithm using $\alpha J$ (which is still anti-symmetric), where $\alpha$ is a constant that can be tuned and it represents the magnitude of non-reversible purturbations. For example, one can choose $J$ to be a circulant matrix, e.g.,

$$-J\nabla F(x) = \left(\partial_{x_d} F(x) - \partial_{x_2} F(x), \partial_{x_1} F(x) - \partial_{x_3} F(x), \ldots, \partial_{x_{d-1}} F(x) - \partial_{x_1} F(x)\right),$$

and this product is easy to implement by shifting the gradient vector in the memory by one unit to the left and one unit to the right and then taking the difference.

## 2 Exit time for non-reversible Langevin dynamics

For convergence to a small neighborhood of the global minimum, Langevin trajectory needs to not only escape from the neighborhood of a local optimum but also exit the basin of attraction of the current minimum and transit to the basin of attraction of other local minima including the global minima. In particular, the convergence rate to a global minimum is controlled by the mean *exit time* from the basin of attraction of a local minima in a potential landscape $F(\cdot)$ in (0.2). In this section we will show that non-reversible Langevin dynamics can lead to faster (smaller) exit times.

Throughout this section, we consider a double-well potential $F : \mathbb{R}^d \to \mathbb{R}$, which has two local minima $a_1, a_2$ with $F(a_2) < F(a_1)$. The two minima are separated by a saddle point $\sigma$. See Figure A in the Supplementary File. In addition to Assumption 1 (i)-(iii), we make generic assumptions that $F \in C^3$, the Hessian of $F$ at each of the local minima is positive definite, and that the Hessian of $F$ at the saddle point $\sigma$ has exactly one strictly negative eigenvalue (denoted as $-\mu^*(\sigma) < 0$) and other eigenvalues are all positive. For the overdamped Langevin diffusion, it is known ([BGK04, Ber13]) that the expected time of the process starting from $a_1$ and hitting a small neighborhood of $a_2$ is:

$$\mathbb{E}\left[\theta_{a_1 \to a_2}^\beta\right] = [1 + o_\beta(1)] \cdot \frac{2\pi}{\mu^*(\sigma)} \cdot e^{\beta[F(\sigma)-F(a_1)]} \cdot \sqrt{\frac{|\det \text{Hess } F(\sigma)|}{\det \text{Hess } F(a_1)}}. \qquad (2.1)$$

Here, $o_\beta(1) \to 0$ as $\beta \to \infty$, $\det \text{Hess } F(x)$ stands for the determinant of the Hessian of $F$ at $x$, and $-\mu^*(\sigma)$ is the unique negative eigenvalue of the Hessian of $F$ at the saddle point $\sigma$. This formula is known as the Eyring-Kramers formula for reversible diffusions. Its rigorous proof was first obtained by [BGK04] by a potential analysis approach, and then by [HKN04] through Witten Laplacian analysis. We refer to [Ber13] for a survey on mathematical approaches to the Eyring-Kramers formula. We note that in many practical applications, for instance in the training of neural networks, the eigenvalues of the Hessian at local extrema concentrate around zero and the magnitude of the eigenvalues $m$ and $\mu^*(\sigma)$ can often be very small (see e.g. [SBL16, CCS+17]).

### 2.1 Underdamped Langevin dynamics

Denote $\Theta_{a_1 \to a_2}^\beta$ as the first time of that the underdamped diffusion (0.4)–(0.5) starting from $a_1$ and hitting a small neighborhood of $a_2$. [BR16] (Remark 5.2) suggests that the expected exit time is:

$$\mathbb{E}\left[\Theta_{a_1 \to a_2}^\beta\right] = [1 + o_\beta(1)] \cdot \frac{2\pi}{\mu_*} \cdot e^{\beta[F(\sigma)-F(a_1)]} \cdot \sqrt{\frac{|\det \text{Hess } F(\sigma)|}{\det \text{Hess } F(a_1)}},$$

where $\mu_*$ is the unique positive eigenvalue of the matrix $\hat{H}_\gamma(\sigma) = \begin{bmatrix} -\gamma I & -\mathbb{L}^\sigma \\ I & 0 \end{bmatrix}$, where $\mathbb{L}^\sigma$ is the Hessian matrix of $F$ at the saddle point $\sigma$. One can readily show that $\mu_*$ is given by the positive eigenvalue of the $2 \times 2$ matrix $\begin{bmatrix} -\gamma & \mu^*(\sigma) \\ 1 & 0 \end{bmatrix}$, which implies that $\mu_* = \frac{1}{2} \cdot \left( \sqrt{\gamma^2 + 4\mu^*(\sigma)} - \gamma \right)$. So if $\gamma + \mu^*(\sigma) < 1$, then we have $\mu_* > \mu^*(\sigma)$ and therefore

$$\lim_{\beta \to \infty} \mathbb{E}\left[ \Theta_{a_1 \to a_2}^\beta \right] \Big/ \mathbb{E}\left[ \theta_{a_1 \to a_2}^\beta \right] = \mu^*(\sigma)/\mu_* < 1. \tag{2.2}$$

That is, the mean exit time for the underdamped diffusion is smaller compared with that of the overdamped diffusion. Roughly speaking, the condition $\gamma + \mu^*(\sigma) < 1$ says that if the curvature of the saddle point in the negative descent direction is not too steep (i.e. if $\mu^*(\sigma) < 1$), we can choose $\gamma$ small enough to accelerate the exit time of the reversible Langevin dynamics. Intuitively speaking, it can be argued that the underdamped process can climb hills and explore the state space faster as it is less likely to go back to the recent states visited due to the momentum term (see e.g. [BR16]). For the discrete time dynamics, it is intuitive to expect that the exit time of the underdamped discrete dynamics is close to that of the continuous time diffusion when the step size is small [BGG17], and hence a similar result as (2.2) will hold for the discrete dynamics when $\gamma + \mu^*(\sigma) < 1$.

To apply the results in [BGG17], we consider a sequence of bounded domains $D_n$ indexed by $n$ so that the following conditions hold: first, the region $D_n$ contains $a_1, a_2, \sigma$ for large $n$; second, as $n$ increases, $D_n$ increases to the set $D_\infty = O^c(a_2) := \mathbb{R}^d \backslash O(a_2)$, where $O(a_2)$ denotes a small neighborhood of $a_2$; third, the underdamped SDE (diffusion) is non-degenerate along the normal direction to the boundary of $D_n$ with probability one. Fix the parameters $\beta$ and $\gamma$ in the underdamped Langevin dynamics. Denote $\hat{\Theta}_{a_1 \to a_2}^{\beta,n}$ be the exit time of $X_k$ (from the ULD dynamics) starting from $a_1$ and exiting domain $D_n$. Fix $\epsilon > 0$. One can choose a sufficiently large $n$ and choose a constant $\tilde{\eta}(\epsilon, n, \gamma, \beta)$ so that for stepsize $\eta \leq \tilde{\eta}(\epsilon, n, \gamma, \beta)$, we have

$$\left| \mathbb{E}\left[ \hat{\Theta}_{a_1 \to a_2}^{\beta,n} \right] - \mathbb{E}\left[ \Theta_{a_1 \to a_2}^\beta \right] \right| < 2\epsilon. \tag{2.3}$$

To see this, we use Theorems 3.9 and 3.11 in [BGG17]. Write $\hat{\theta}_{a_1 \to a_2}^{\beta,n}$ as the exit time of $X_k$ (from the overdamped discrete dynamics) starting from $a_1$ and exiting the domain $D_n$. Then one can also expect that when $n$ is large and the step size is small, the mean of $\hat{\theta}_{a_1 \to a_2}^{\beta,n}$ will be close to $\mathbb{E}\left[ \theta_{a_1 \to a_2}^\beta \right]$, the continuous exit time of the overdamped diffusion given in (2.1). See Proposition 6 in the next section. It then follows from (2.2) and (2.3) that for large enough $\beta, n$, and sufficiently small stepsize $\eta$, we obtain the acceleration in discrete time:

$$\mathbb{E}\left[ \hat{\Theta}_{a_1 \to a_2}^{\beta,n} \right] \Big/ \mathbb{E}\left[ \hat{\theta}_{a_1 \to a_2}^{\beta,n} \right] = \mathcal{O}\left( \sqrt{\mu^*(\sigma)} \right) < 1.$$

## 2.2 Non-reversible Langevin dynamics

[LMS19] (Theorem 5.2) showed that the expected time of the non-reversible diffusion $X(t)$ in (0.9) starting from $a_1$ and hitting a small neighborhood of $a_2$ is given by

$$\mathbb{E}\left[ \tau_{a_1 \to a_2}^\beta \right] = [1 + o_\beta(1)] \cdot \frac{2\pi}{\mu_J^*} \cdot e^{\beta[F(\sigma) - F(a_1)]} \cdot \sqrt{\frac{|\det \operatorname{Hess} F(\sigma)|}{\det \operatorname{Hess} F(a_1)}}. \tag{2.4}$$

Here, $-\mu_J^*$ is the unique negative eigenvalue of the matrix $A_J \cdot \mathbb{L}^\sigma$, where $\mathbb{L}^\sigma := \operatorname{Hess} F(\sigma)$, the Hessian of $F$ at the saddle point $\sigma$. We denote $u$ for the corresponding eigenvector of $A_J \mathbb{L}^\sigma$ for the eigenvalue $-\mu_J^* < 0$. and $\mathbb{L}^\sigma = S^T D S$, for a real orthogonal matrix $S$, where $D = \operatorname{diag}(\mu_1, \mu_2, \ldots, \mu_d)$ with $\mu_1 < 0 < \mu_2 < \ldots < \mu_d$ being the eigenvalues of $\mathbb{L}^\sigma$.

**Proposition 5.** *We have $\mu_J^* \geq \mu^*(\sigma)$. As a consequence,*

$$\lim_{\beta \to \infty} \mathbb{E}\left[ \tau_{a_1 \to a_2}^\beta \right] \Big/ \mathbb{E}\left[ \tau_{a_1 \to a_2}^\beta \right]_{J=0} = \mu^*(\sigma)/\mu_J^* \leq 1. \tag{2.5}$$

*The equality is attained if and only if $u$ is a singular vector of $J$ satisfying $Ju = 0$.*

Proposition 5 shows that if $J$ is not singular, the non-reversible dynamics is generically faster than the reversible dynamics in the sense of smaller mean exit times. This holds for the discrete dynamics

as well, since the exit time for the discrete dynamics is close to that of the continuous dynamics for sufficiently small stepsizes, see e.g. [GM05].

Next let us discuss the discrete dynamics (0.10). Unlike the underdamped diffusion which has a non-invertible diffusion matrix, the non-reversible Langevin diffusion in (0.9) is uniformly elliptic. So to show the discrete exit time is close to the continuous exit time for non-reversible Langevin dynamics, we can apply the results in [GM05] and proceed as follows. Let $B_n$ be the ball centered at zero with radius $n$ in $\mathbb{R}^d$. For $n$ sufficiently large, we always have $a_1, a_2, \sigma \in B_n$. Let $\bar{D}_n = B_n \setminus O(a_2)$, where $O(a_2)$ denotes a small neighborhood of $a_2$. It follows that the set $\bar{D}_n$ is bounded for each $n$, and it increases to the set $O^c(a_2)$ as $n$ is sent to infinity. Write $\hat{\tau}^{\beta,n}_{a_1 \to a_2}$ for the first time that the discrete-time dynamics starting from $a_1$ and exit the region $\bar{D}_n$. Then we can obtain from [GM05] the following result, which implies the exit times of non-reversible Langevin dynamics is smaller compared with that of reversible Langevin dynamics.

**Proposition 6.** *Fix the antisymmetric matrix $J$, the temperature parameter $\beta$, and $\epsilon > 0$. One can choose a sufficiently large $n$ and a constant $\bar{\eta}(\epsilon, n, \beta)$ so that for stepsize $\eta \leq \bar{\eta}(\epsilon, n, \beta)$, we have*

$$\left| \mathbb{E}\left[\hat{\tau}^{\beta,n}_{a_1 \to a_2}\right] - \mathbb{E}\left[\tau^{\beta}_{a_1 \to a_2}\right] \right| < 2\epsilon.$$

*It then follows from Proposition 5 that for large $\beta$ we have*

$$\mathbb{E}\left[\hat{\tau}^{\beta,n}_{a_1 \to a_2}\right] \Big/ \mathbb{E}\left[\hat{\tau}^{\beta,n}_{a_1 \to a_2}\right]_{J=0} < 1, \tag{2.6}$$

*provided that $(Su)_i \neq 0$ for some $i \in \{2, \dots, d\}$ which occurs if and only if $u$ is a singular vector of $J$ satisfying $Ju = 0$.*

## 3  Numerical Illustrations

**Choice of algorithm parameters.** In practice, for the NLD algorithm, the matrix $J$ can be chosen as a random anti-symmetric matrix. For quadratic objectives, there is a formula for optimal $J$ matrix; see e.g. [LNP13]. For the ULD algorithm, we can take the parameter $\gamma = 2\sqrt{m}$ as predicted by our theory (Lemma 2) for quadratics. On the top panel of Figure 1, we compare ULD and NLD to LD for the double well example with random initialization over 100 runs where $J$ is chosen randomly and $\gamma = 2\sqrt{m}$. In this simple example, we observe NLD and ULD have smaller mean exit times (from a barrier) compared to LD.

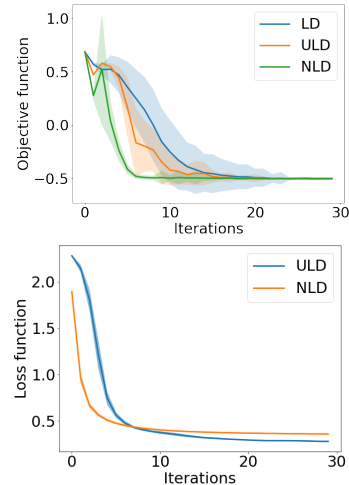

Figure 1: Choice of algorithm parameters and comparing ULD and NLD.

**Comparing ULD and NLD algorithms.** In general, it is not easy to compare the theoretical performance of ULD and NLD algorithms. However, in some regimes, our theory predicts one is better than the other. For example, when the smallest eigenvalue $m$ is close to the largest eigenvalue $M$, NLD will not improve much upon LD, but ULD will improve upon LD if $m$ is small and ULD will be faster than NLD. In this case, ULD has better performance than NLD. On the bottom panel of Figure 1, we provide an example for training fully-connected neural networks on MNIST where ULD was faster when both methods were tuned.

## 4  Conclusion

Langevin Monte Carlo are powerful tools for sampling from a target distribution as well as for optimizing a non-convex objective. The classic Langevin dynamics (LD) is based on the first-order Langevin diffusion which is reversible in time. We studied the two variants that are based on non-reversible Langevin diffusions: the underdamped Langevin dynamics (ULD) and the Langevin dynamics with a non-symmetric drift (NLD). We showed that both ULD and NLD can improve upon the recurrence time for LD in [TLR18] and discussed the amount of improvement. We also showed that non-reversible variants can exit the basin of attraction of a local minimimum faster when the objective has two local minima separated by a saddle point and discussed the amount of improvement. By breaking the reversibility in the Langevin dynamics, our results quantify the improvement in performance and fill a gap between the theory and practice of non-reversible Langevin algorithms.

## Broader Impact

Langevin algorithms are core Markov Chain Monte Carlo (MCMC) methods for solving machine learning problems. These methods arise in several contexts in machine learning and data science. For example, they can be applied to Bayesian inference problems. They can also be used to solve stochastic non-convex optimization problems including the challenging problems arising in deep learning. Our paper argues that the non-reversible variants of the classical Langevin algorithms can perform better by providing rigorous mathematical analysis, and bridges a gap between theory and practice. Therefore, our paper contributes to the growing literature on theoretical foundations of MCMC methods. Researchers in the machine learning community and beyond will benefit from this research by having a better understanding of why non-reversible variants of the classical Langevin algorithms can improve performance.

## Acknowledgements and Disclosure of Funding

The authors thank four anonymous referees for helpful suggestions. The authors are very grateful to Emmanuel Gobet, Claudio Landim, Insuk Seo and Gabriel Stoltz for helpful comments and discussions. The authors also thank Yuanhan Hu for the help with numerical experiments. Xuefeng Gao acknowledges support from Hong Kong RGC Grants 24207015 and 14201117. Mert Gürbüzbalaban's research is supported in part by the grants NSF DMS-1723085 and NSF CCF-1814888. Lingjiong Zhu is grateful to the support from the grant NSF DMS-1613164.

## Footnotes

[2]This algorithm is also known as the kinetic Langevin Monte Carlo algorithm.

[3]Here, we abuse the notation, and for simplicity we use the same letter $m$ to denote the smallest eigenvalue in (1.2) and in the dissipativity constant in Assumption 1 (iii). If these two constants are different, the constant $m$ in our main results can be taken as the former, i.e. the smallest eigenvalue of the Hessian at $x_*$.

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
