[Supplementary Material]

# A Figures

Figure 1: The norm $\|e^{-tH_\gamma}\|$ is optimized for the choice of $\gamma = 2\sqrt{m}$. This is illustrated in the figure for $m = 0.01$.

Figure 2: A double-well example. Here, $\Delta F = F(\sigma) - F(a_1)$. There are exactly two local minima $a_1$ and $a_2$ which are separated with a saddle point $\sigma$.

# B Proof of results in Section 2

## B.1 Proof of Lemma 2

*Proof.* Let $H$ be a symmetric positive definite matrix with eigenvalue decomposition $H = QDQ^T$, where $D$ is diagonal with eigenvalues in increasing order $m := \lambda_1 \le \lambda_2 \le \cdots \le \lambda_d =: M$ of the matrix $H$. Recall $H_\gamma$ from (2.2). Note that

$$H_\gamma = \begin{bmatrix} Q & 0 \\ 0 & Q \end{bmatrix} G_\gamma \begin{bmatrix} Q^T & 0 \\ 0 & Q^T \end{bmatrix}, \quad G_\gamma := \begin{bmatrix} \gamma I & D \\ -I & 0 \end{bmatrix}.$$

Therefore $H_\gamma$ and $G_\gamma$ have the same eigenvalues. Due to the structure of $G_\gamma$, it can be seen that there exists a permutation matrix $P$ such that

$$T_\gamma := PG_\gamma P^T = \begin{bmatrix} T_1(\gamma) & 0 & 0 & 0 \\ 0 & T_2(\gamma) & 0 & 0 \\ \vdots & \cdots & \ddots & \vdots \\ 0 & 0 & 0 & T_d(\gamma) \end{bmatrix}, \quad \text{where} \quad T_i(\gamma) := \begin{bmatrix} \gamma & \lambda_i \\ -1 & 0 \end{bmatrix}, \quad \text{(B.1)}$$

with $i = 1, 2, \ldots, d$, and $T_i(\gamma)$ are $2 \times 2$ block matrices with the eigenvalues:

$$\mu_{i,\pm} := \frac{\gamma \pm \sqrt{\gamma^2 - 4\lambda_i}}{2}. \quad i = 1, 2, \ldots, d. \quad \text{(B.2)}$$

448  We observe that $T_\gamma$ and $G_\gamma$ (and therefore $H_\gamma$) have the same eigenvalues and the eigenvalues of $T_\gamma$
449  are determined by the eigenvalues of the $2 \times 2$ block matrices $T_i(\gamma)$.

450  Since $H_\gamma$ is unitarily equivalent to the matrix $T_\gamma$, i.e. there exists a unitary matrix $U$ such that
451  $H_\gamma = U T_\gamma U^*$, we have $\left\| e^{-t H_\gamma} \right\| = \left\| U e^{-t T_\gamma} U^* \right\| = \left\| e^{-t T_\gamma} \right\|$. Since $T_\gamma$ is a block diagonal matrix
452  with $2 \times 2$ blocks $T_i(\gamma)$ we have $\left\| e^{-t T_\gamma} \right\| = \max_{1 \le i \le d} \left\| e^{-t T_i(\gamma)} \right\|$. Assume that $\gamma^2 - 4\lambda_1 =$
453  $\gamma^2 - 4m \le 0$ so that the eigenvalues $\mu_{i,\pm}$ of $T_i(\gamma)$ (see Eqn. (B.2)) are real when $\gamma = 2\sqrt{m}$ and
454  complex when $\lambda < 2\sqrt{m}$. Note that

$$\left\| e^{-t T_i(\gamma)} \right\| = e^{-t\gamma/2} \left\| e^{-t \tilde{T}_i(\gamma)} \right\|, \qquad \text{where} \quad \tilde{T}_i(\gamma) := T_i(\gamma) - \frac{\gamma}{2} I, \qquad 1 \le i \le d. \quad \text{(B.3)}$$

455  We consider $\gamma \in (0, 2\sqrt{m}]$. Depending on the value of $\lambda_i$ and $\gamma$, there are two cases:

456  **Case 1**. If $\gamma < 2\sqrt{m}$ or ($\lambda_i > m$ and $\gamma = 2\sqrt{m}$), then $\tilde{T}_i(\gamma)$ has purely imaginary eigenvalues
457  that are complex conjugates which we denote by $\tilde{\mu}_{i,\pm} = \pm i \frac{\sqrt{4\lambda_i - \gamma^2}}{2}$, $1 \le i \le d$. We will show
458  that the last term in (B.3) stays bounded due to the imaginariness of the eigenvalues of $\tilde{T}_i(\gamma)$. It
459  is easy to check that $2 \times 2$ matrix $\tilde{T}_i(\gamma)$ have the eigenvectors $v_{i,\pm} = [\mu_{i,\pm}, -1]^T$. If we set
460  $G_i := [v_{i,+} \quad v_{i,-}] \in \mathbb{C}^{2 \times 2}$, the eigenvalue decomposition of $\tilde{T}_i(\gamma)$ is given by

$$\tilde{T}_i(\gamma) = G_i \begin{bmatrix} \tilde{\mu}_{i,+} & 0 \\ 0 & \tilde{\mu}_{i,-} \end{bmatrix} G_i^{-1}, \quad \text{where} \quad G_i^{-1} = \frac{1}{\det G_i} \begin{bmatrix} -1 & -\mu_{i,-} \\ 1 & \mu_{i,+} \end{bmatrix},$$

461  and $\det G_i = i \sqrt{4\lambda_i - \gamma^2}$. We can compute that

$$
\begin{aligned}
e^{-t \tilde{T}_i(\gamma)} &= G_i \begin{bmatrix} e^{-it\sqrt{4\lambda_i - \gamma^2}/2} & 0 \\ 0 & e^{it\sqrt{4\lambda_i - \gamma^2}/2} \end{bmatrix} G_i^{-1} \\
&= \frac{1}{\det G_i} \begin{bmatrix} \mu_{i,+} & \mu_{i,-} \\ -1 & -1 \end{bmatrix} \begin{bmatrix} -e^{-it\sqrt{4\lambda_i - \gamma^2}/2} & -\mu_{i,-} e^{-it\sqrt{4\lambda_i - \gamma^2}/2} \\ e^{it\sqrt{4\lambda_i - \gamma^2}/2} & \mu_{i,+} e^{it\sqrt{4\lambda_i - \gamma^2}/2} \end{bmatrix} \\
&= \frac{1}{i\sqrt{4\lambda_i - \gamma^2}} \begin{bmatrix} 2\mathrm{Imag}\left(\mu_{i,-} e^{it\sqrt{4\lambda_i - \gamma^2}/2}\right) & 2i|\mu_{i,+}|^2 \sin\left(t\sqrt{4\lambda_i - \gamma^2}/2\right) \\ -2i \sin\left(t\sqrt{4\lambda_i - \gamma^2}/2\right) & 2\mathrm{Imag}\left(\mu_{i,+} e^{it\sqrt{4\lambda_i - \gamma^2}/2}\right) \end{bmatrix},
\end{aligned}
$$

462  where $\mathrm{Imag}(a + ib) := ib$ denotes the imaginary part of a complex number. As a consequence, by
463  taking componentwise absolute values

$$
\begin{aligned}
\left\| e^{-t \tilde{T}_i(\gamma)} \right\| &\le \frac{1}{\sqrt{4\lambda_i - \gamma^2}} \left\| \begin{bmatrix} 2|\mu_{i,-}| & 2|\mu_{i,+}|^2 \\ 2 & 2|\mu_{i,+}| \end{bmatrix} \right\| = \frac{1}{\sqrt{4\lambda_i - \gamma^2}} \left\| \begin{bmatrix} 2\sqrt{\lambda_i} & 2\lambda_i \\ 2 & 2\sqrt{\lambda_i} \end{bmatrix} \right\| \\
&= \frac{1}{\sqrt{4\lambda_i - \gamma^2}} \left\| \begin{bmatrix} 2\sqrt{\lambda_i} \\ 2 \end{bmatrix} \begin{bmatrix} 1 & \sqrt{\lambda_i} \end{bmatrix} \right\| = \frac{1}{\sqrt{4\lambda_i - \gamma^2}} \left\| \begin{bmatrix} 2\sqrt{\lambda_i} \\ 2 \end{bmatrix} \right\| \left\| \begin{bmatrix} 1 & \sqrt{\lambda_i} \end{bmatrix} \right\| \\
&= \frac{2(1 + \lambda_i)}{\sqrt{4\lambda_i - \gamma^2}}, \quad \text{(B.4)}
\end{aligned}
$$

464  where the second from last equality used the fact that the 2-norm of a rank-one matrix is equal
465  to its Frobenius norm. [2] Then, it follows from (B.3) that $\left\| e^{-t T_i(\gamma)} \right\| = e^{-t\gamma/2} \left\| e^{-t \tilde{T}_i(\gamma)} \right\| \le$
466  $\frac{2(1+\lambda_i)}{\sqrt{4\lambda_i - \gamma^2}} e^{-t\gamma/2}$, which implies $\left\| e^{-t H_\gamma} \right\| = \left\| e^{-t T_\gamma} \right\| \le \max_{1 \le i \le d} \left\| e^{-t T_i(\gamma)} \right\| \le \frac{2(1+M)}{\sqrt{4m - \gamma^2}} e^{-t\gamma/2}$,
467  provided that $\gamma^2 - 4m < 0$. In particular, if we choose $\hat{\varepsilon} = 1 - \frac{\gamma}{2\sqrt{m}}$ for any $\hat{\varepsilon} > 0$, we obtain

$$\left\| e^{-t H_\gamma} \right\| \le \frac{1 + M}{\sqrt{m(1 - (1 - \hat{\varepsilon})^2)}} e^{-\sqrt{m}(1-\hat{\varepsilon})t}.$$

468 The proof for **Case 1** is complete.

**Case 2**. If $\gamma = 2\sqrt{m}$ and $\lambda_i = m$, then $\tilde{T}_i(\gamma)$ has double eigenvalues at zero and is not diagonalizable. It admits the Jordan decomposition

$$\tilde{T}_i(\gamma) = G_i \begin{bmatrix} 0 & 1 \\ 0 & 0 \end{bmatrix} G_i^{-1} \quad \text{with} \quad G_i = \begin{bmatrix} \sqrt{m} & 1 \\ -1 & 0 \end{bmatrix} \quad \text{and} \quad G_i^{-1} = \begin{bmatrix} 0 & -1 \\ 1 & \sqrt{m} \end{bmatrix}.$$

By a direct computation, we obtain

$$e^{-t\tilde{T}_i(\gamma)} = G_i \begin{bmatrix} 1 & -t \\ 0 & 1 \end{bmatrix} G_i^{-1} = \begin{bmatrix} 1 - t\sqrt{m} & -tm \\ t & 1 + t\sqrt{m} \end{bmatrix}.$$

469 A simple computation reveals

$$\left\| e^{-t\tilde{T}_i(\gamma)} \right\| \le \sqrt{\mathrm{Tr}\left( e^{-t\tilde{T}_i(\gamma)} e^{-t\tilde{T}_i(\gamma)^T} \right)} = \sqrt{2 + (m+1)^2 t^2}. \tag{B.5}$$

470 To finish the proof of **Case 2**, let $\gamma = 2\sqrt{m}$. We compute

$$\max_{1 \le i \le d} \left\| e^{-t\tilde{T}_i(\gamma)} \right\| = \max \left\{ \max_{i:\lambda_i = m} \left\| e^{-t\tilde{T}_i(\gamma)} \right\|, \max_{i:\lambda_i > m} \left\| e^{-t\tilde{T}_i(\gamma)} \right\| \right\}$$

$$\le \max \left\{ \sqrt{2 + (m+1)^2 t^2}, \max_{i:\lambda_i > m} \frac{(1 + \lambda_i)}{\sqrt{\lambda_i - m}} \right\},$$

471 where we used (B.4) and (B.5) in the last inequality. We conclude from (B.3) for **Case 2**. □

## B.2 Proof of Theorem 3

473 The main result we use to prove Theorem 3 is the following proposition. The proof of the following
474 result will be presented later in Section B.2.2.

475 **Proposition 7.** *Assume $\gamma = 2\sqrt{m}$. Fix any $r > 0$ and*

$$0 < \varepsilon < \min \left\{ \bar{\varepsilon}_1^U, \bar{\varepsilon}_2^U, \bar{\varepsilon}_3^U \right\},$$

476 *where*

$$\bar{\varepsilon}_1^U := \sqrt{\frac{C_H + 2 + (m+1)^2}{(C_H + 2)m + (m+1)^2}} r, \tag{B.6}$$

$$\bar{\varepsilon}_2^U := 2\sqrt{2} \left( C_H + 2 + (m+1)^2 \right)^{1/4} \frac{e^{-1/2} r}{m^{1/4}}, \tag{B.7}$$

$$\bar{\varepsilon}_3^U := \frac{\sqrt{m}}{4L \left( \sqrt{C_H + 2} + \frac{m+1}{\sqrt{m}} + \frac{\sqrt{(C_H + 2)m + (m+1)}}{8\sqrt{C_H + 2 + (m+1)^2}} \right)}. \tag{B.8}$$

477 *Consider the stopping time:*

$$\tau := \inf \left\{ t \ge 0 : \|X(t) - x_*\| \ge \varepsilon + r e^{-\sqrt{m}t} \right\}.$$

478 *For any initial point $X(0) = x$ with $\|x - x_*\| \le r$, and*

$$\beta \ge \frac{256(2C_H m + 4m + (m+1)^2)}{m\varepsilon^2} \left( d\log(2) + \log\left( \frac{2\|H_{2\sqrt{m}}\|\mathcal{T} + 1}{\delta} \right) \right),$$

479 *we have*

$$\mathbb{P}_x \left( \tau \in [\mathcal{T}_{rec}^U, \mathcal{T}_{esc}^U] \right) \le \delta.$$

480 We are now ready to complete the proof of Theorem 3.

### B.2.1 Completing the proof of Theorem 3

Assume that $\gamma = 2\sqrt{m}$. Let us compare the discrete dynamics (1.7)-(1.8) and the continuous dynamics (1.4)-(1.5). Define:

$$\tilde{V}(t) = V_0 - \int_0^t \gamma \tilde{V}\left(\lfloor s/\eta \rfloor \eta\right) ds - \int_0^t \nabla F\left(\tilde{X}\left(\lfloor s/\eta \rfloor \eta\right)\right) ds + \sqrt{2\gamma\beta^{-1}} \int_0^t dB_s, \qquad \text{(B.9)}$$

$$\tilde{X}(t) = X_0 + \int_0^t \tilde{V}\left(\lfloor s/\eta \rfloor \eta\right) ds. \qquad \text{(B.10)}$$

The process $(\tilde{V}, \tilde{X})$ defined in (B.9) and (B.10) is the continuous-time interpolation of the iterates $\{(V_k, X_k)\}$. In particular, the joint distribution of $\{(V_k, X_k) : k = 1, 2, \ldots, K\}$ is the same as $\{(\tilde{V}(t), \tilde{X}(t)) : t = \eta, 2\eta, \ldots, K\eta\}$ for any positive integer $K$.

It is derived in the proof of Lemma EC.6 in [GGZ18] that the relative entropy $D(\cdot\|\cdot)$ between the law $\tilde{\mathbb{P}}^{K\eta}$ of $((\tilde{V}(t), \tilde{X}(t)) : t \leq K\eta)$ and the law $\mathbb{P}^{K\eta}$ of $((V(t), X(t)) : t \leq K\eta)$ is upper bounded as follows:

$$D\left(\tilde{\mathbb{P}}^{K\eta} \middle\| \mathbb{P}^{K\eta}\right) \leq \frac{3\beta M^2}{2\gamma} K\eta^3 \left(C_v^d + 2M^2 C_x^d + 2B^2 + \frac{2d\gamma\beta^{-1}}{3}\right),$$

provided that $\eta \leq \min\left\{1, \frac{\gamma}{\tilde{K}_2}(d/\beta + \overline{A}/\beta), \frac{\gamma\lambda}{2\tilde{K}_1}\right\}$, where $C_v^d$ is defined in Lemma 10. Using Pinsker's inequality, we obtain an upper bound on the total variation $\|\cdot\|_{TV}$:

$$\left\|\tilde{\mathbb{P}}^{K\eta} - \mathbb{P}^{K\eta}\right\|_{TV}^2 \leq \frac{3\beta M^2}{4\gamma} K\eta^3 \left(C_v^d + 2M^2 C_x^d + 2B^2 + \frac{2d\gamma\beta^{-1}}{3}\right).$$

Using a result about an optimal coupling (Theorem 5.2., [Lin92]), that is, given any two random elements $\mathcal{X}, \mathcal{Y}$ of a common standard Borel space, there exists a coupling $\mathcal{P}$ of $\mathcal{X}$ and $\mathcal{Y}$ such that

$$\mathcal{P}(\mathcal{X} \neq \mathcal{Y}) \leq \|\mathcal{L}(\mathcal{X}) - \mathcal{L}(\mathcal{Y})\|_{TV}.$$

Hence, given any $\beta > 0$ and $K\eta \leq \mathcal{T}_{\text{esc}}^U$, we can choose

$$\eta^2 \leq \frac{4\gamma\delta^2}{3\beta M^2(C_v^d + 2M^2 C_x^d + 2B^2 + \frac{2d\gamma\beta^{-1}}{3})\mathcal{T}_{\text{esc}}^U}, \qquad \text{(B.11)}$$

so that there is a coupling of $\{(V(k\eta), X(k\eta)) : k = 1, 2, \ldots, K\}$ and $\{(V_k, X_k) : k = 1, 2, \ldots, K\}$ such that

$$\mathcal{P}(((V(\eta), X(\eta)), \ldots, (V(K\eta), X(K\eta))) \neq ((V_1, X_1), \ldots, (V_K, X_K)) \leq \delta. \qquad \text{(B.12)}$$

It follows that

$$\mathbb{P}(((V_1, X_1), \ldots, (V_K, X_K)) \in \cdot) \leq \mathbb{P}(((V(\eta), X(\eta)), \ldots, (V(K\eta), X(K\eta))) \in \cdot) + \delta.$$

Let us now complete the proof of Theorem 3. We need to show that

$$\mathbb{P}\left((X_1, \ldots, X_K) \in \mathcal{A}\right) \leq \delta,$$

where $K = \lfloor \eta^{-1} \mathcal{T}_{\text{esc}}^U \rfloor$ and $\mathcal{A} := \mathcal{A}_1 \cap \mathcal{A}_2$, where

$$\mathcal{A}_1 := \left\{(x_1, \ldots, x_K) \in (\mathbb{R}^d)^K : \max_{k \leq \eta^{-1}\mathcal{T}_{\text{rec}}^U} \frac{\|x_k - x_*\|}{\varepsilon + re^{-\sqrt{m}k\eta}} \leq \frac{1}{2}\right\},$$

$$\mathcal{A}_2 := \left\{(x_1, \ldots, x_K) \in (\mathbb{R}^d)^K : \max_{\eta^{-1}\mathcal{T}_{\text{rec}}^U \leq k \leq K} \frac{\|x_k - x_*\|}{\varepsilon + re^{-\sqrt{m}k\eta}} \geq 1\right\}.$$

We can choose $\beta$ sufficiently large so that with probability at least $1 - \delta/3$, we have either $\|X(t) - x_*\| \geq \varepsilon + re^{-\sqrt{m}t}$ for some $t \leq \mathcal{T}_{\text{rec}}^U$ or $\|X(t) - x_*\| \leq \varepsilon + re^{-\sqrt{m}t}$ for all $t \leq \mathcal{T}_{\text{esc}}^U$. Moreover, for any $K, \eta$ and $\beta$ satisfying the conditions of the theorem, there exists a coupling of $(X(\eta), \ldots, X(K\eta))$ and $(X_1, \ldots, X_K)$ so that with probability $1 - \delta/3$, $X_k = X(k\eta)$ for all $k = 1, 2, \ldots, K$. Then, by (B.11) and (B.12), we get

$$\mathbb{P}((X_1, \ldots, X_K) \in \mathcal{A}) \leq \mathbb{P}((X(\eta), \ldots, X(K\eta)) \in \mathcal{A}) + \frac{\delta}{3}, \qquad \text{(B.13)}$$

provided that

$$\eta \leq \overline{\eta}_3^U := \frac{2\gamma^{1/2}\delta}{3\sqrt{3\beta}M(C_v^d + 2M^2C_x^d + 2B^2 + \frac{2d\gamma\beta^{-1}}{3})^{1/2}(\mathcal{T}_{\text{esc}}^U)^{1/2}}. \tag{B.14}$$

It remains to estimate the probability of $\mathbb{P}((X(\eta), \ldots, X(K\eta)) \in \mathcal{A}_1 \cap \mathcal{A}_2)$ for the underdamped Langevin diffusion. Partition the interval $[0, \mathcal{T}_{\text{rec}}^U]$ using the points $0 = t_1 < t_1 < \cdots < t_{\lceil \eta^{-1}\mathcal{T}_{\text{rec}}^U \rceil} = \mathcal{T}_{\text{rec}}^U$ with $t_k = k\eta$ for $k = 0, 1, \ldots, \lceil \eta^{-1}\mathcal{T}_{\text{rec}}^U \rceil - 1$, and consider the event:

$$\mathcal{B} := \left\{ \max_{0 \leq k \leq \lceil \eta^{-1}\mathcal{T}_{\text{rec}}^U \rceil - 1} \max_{t \in [t_k, t_{k+1}]} \|X(t) - X(t_{k+1})\| \leq \frac{\varepsilon}{2} \right\}.$$

On the event $\{(X(\eta), \ldots, X(K\eta)) \in \mathcal{A}_1\} \cap \mathcal{B}$,

$$\sup_{t \in [0, \mathcal{T}_{\text{rec}}^U]} \frac{\|X(t) - x_*\|}{\varepsilon + re^{-\sqrt{m}t}} = \max_{0 \leq k \leq \lceil \eta^{-1}\mathcal{T}_{\text{rec}}^U \rceil - 1} \sup_{t \in [t_k, t_{k+1}]} \frac{\|X(t) - x_*\|}{\varepsilon + re^{-\sqrt{m}t}}$$

$$\leq \frac{1}{2} + \max_{0 \leq k \leq \lceil \eta^{-1}\mathcal{T}_{\text{rec}}^U \rceil - 1} \max_{t \in [t_k, t_{k+1}]} \frac{1}{\varepsilon} \|X(t) - X(t_{k+1})\| < 1,$$

and thus

$$\mathbb{P}((X(\eta), \cdots, X(K\eta)) \in \mathcal{A}) \leq \mathbb{P}(\{(X(\eta), \cdots, X(K\eta)) \in \mathcal{A}\} \cap \mathcal{B}) + \mathbb{P}(\mathcal{B}^c)$$

$$\leq \mathbb{P}(\tau \in [\mathcal{T}_{\text{rec}}^U, \mathcal{T}_{\text{esc}}^U]) + \mathbb{P}(\mathcal{B}^c)$$

$$\leq \frac{\delta}{3} + \mathbb{P}(\mathcal{B}^c), \tag{B.15}$$

provided that (by applying Proposition 7 and Lemma 18) (with $\gamma = 2\sqrt{m}$):

$$\beta \geq \underline{\beta}_1^U := \frac{256(2C_H m + 4m + (m+1)^2)}{m\varepsilon^2} \left( d\log(2) + \log\left( \frac{6\sqrt{4m + M^2 + 1}\mathcal{T} + 3}{\delta} \right) \right). \tag{B.16}$$

To complete the proof, we need to show that $\mathbb{P}(\mathcal{B}^c) \leq \frac{\delta}{3}$ in view of (B.13) and (B.15). For any $t \in [t_k, t_{k+1}]$, where $t_{k+1} - t_k = \eta$, we have

$$\|X(t) - X(t_{k+1})\| \leq \int_t^{t_{k+1}} \|V(s)\|ds \leq \eta\|V(t_{k+1})\| + \int_t^{t_{k+1}} \|V(s) - V(t_{k+1})\|ds, \quad \text{(B.17)}$$

and

$$\|V(t) - V(t_{k+1})\|$$

$$\leq \gamma \int_t^{t_{k+1}} \|V(s)\|ds + \int_t^{t_{k+1}} \|\nabla F(X(s))\|ds + \sqrt{2\gamma\beta^{-1}}\|B_t - B_{t_{k+1}}\|$$

$$\leq \gamma\eta\|V(t_{k+1})\| + \gamma \int_t^{t_{k+1}} \|V(s) - V(t_{k+1})\|ds$$

$$+ M \int_t^{t_{k+1}} \|X(s) - X(t_{k+1})\|ds + \eta\|\nabla F(X(t_{k+1}))\| + \sqrt{2\gamma\beta^{-1}}\|B_t - B_{t_{k+1}}\|$$

$$\leq \gamma\eta\|V(t_{k+1})\| + \gamma \int_t^{t_{k+1}} \|V(s) - V(t_{k+1})\|ds$$

$$+ M \int_t^{t_{k+1}} \|X(s) - X(t_{k+1})\|ds + M\eta\|X(t_{k+1})\| + B\eta + \sqrt{2\gamma\beta^{-1}}\|B_t - B_{t_{k+1}}\|, \tag{B.18}$$

where the second inequality above used $M$-Lipschitz property of $\nabla F$ and the last inequality above used Lemma 20. By adding the above two inequalities (B.17) and (B.18) together, we get

$$\|X(t) - X(t_{k+1})\| + \|V(t) - V(t_{k+1})\|$$

$$\leq (1+\gamma)\eta\|V(t_{k+1})\| + (1+\gamma)\int_t^{t_{k+1}} \|V(s) - V(t_{k+1})\|ds$$

$$+ M\int_t^{t_{k+1}} \|X(s) - X(t_{k+1})\|ds + M\eta\|X(t_{k+1})\| + B\eta + \sqrt{2\gamma\beta^{-1}}\|B_t - B_{t_{k+1}}\|$$

$$\leq (1+\gamma+M)\int_t^{t_{k+1}} \left(\|V(s) - V(t_{k+1})\| + \|X(s) - X(t_{k+1})\|\right)ds$$

$$+ (1+\gamma)\eta\|V(t_{k+1})\| + M\eta\|X(t_{k+1})\| + B\eta + \sqrt{2\gamma\beta^{-1}}\sup_{t\in[t_k,t_{k+1}]}\|B_t - B_{t_{k+1}}\|.$$

By applying Gronwall's inequality, we get

$$\sup_{t\in[t_k,t_{k+1}]}\left[\|X(t) - X(t_{k+1})\| + \|V(t) - V(t_{k+1})\|\right]$$

$$\leq e^{(1+\gamma+M)\eta}\left[(1+\gamma)\eta\|V(t_{k+1})\| + M\eta\|X(t_{k+1})\| + B\eta + \sqrt{2\gamma\beta^{-1}}\sup_{t\in[t_k,t_{k+1}]}\|B_t - B_{t_{k+1}}\|\right].$$
$$\tag{B.19}$$

We have from Lemma 10 that for any $u > 0$,

$$\mathbb{P}(\|V(t_{k+1})\| \geq u) \leq \frac{\sup_{t>0}\mathbb{E}\|V(t)\|^2}{u^2} \leq \frac{C_v^c}{u^2}, \tag{B.20}$$

and

$$\mathbb{P}(\|X(t_{k+1})\| \geq u) \leq \frac{\sup_{t>0}\mathbb{E}\|X(t)\|^2}{u^2} \leq \frac{C_x^c}{u^2}, \tag{B.21}$$

where $C_v^c$, $C_x^c$ are defined in Lemma 10. By Lemma 19, we have

$$\mathbb{P}\left(\sup_{t\in[t_k,t_{k+1}]}\|B_t - B_{t_{k+1}}\| \geq u\right) \leq 2^{1/4}e^{1/4}e^{-\frac{u^2}{4d\eta}}.$$

Therefore, we can infer from (B.19) that with $K_0 := \lceil\eta^{-1}\mathcal{T}_{\text{rec}}^U\rceil$,

$$\mathbb{P}(\mathcal{B}^c)$$

$$\leq \sum_{k=0}^{K_0-1}\mathbb{P}\left(\|X(t_{k+1})\| \geq \frac{\varepsilon e^{-(1+\gamma+M)\eta}}{8M\eta}\right) + \sum_{k=0}^{K_0-1}\mathbb{P}\left(\|V(t_{k+1})\| \geq \frac{\varepsilon e^{-(1+\gamma+M)\eta}}{8(1+\gamma)\eta}\right)$$

$$+ \sum_{k=0}^{K_0-1}\mathbb{P}\left(B \geq \frac{\varepsilon e^{-(1+\gamma+M)\eta}}{8\eta}\right) + \sum_{k=0}^{K_0-1}\mathbb{P}\left(\sup_{t\in[t_k,t_{k+1}]}\|B_t - B_{t_{k+1}}\| \geq \frac{\varepsilon e^{-(1+\gamma+M)\eta}\sqrt{\beta}}{8\sqrt{2\gamma}}\right)$$

$$\leq \frac{64K_0}{\varepsilon^2}\left(M^2 C_x^c + (1+\gamma)^2 C_v^c\right)\cdot\eta^2 e^{2(1+\gamma+M)\eta} \tag{B.22}$$

$$+ 2^{1/4}e^{1/4}K_0\cdot\exp\left(-\frac{1}{4d\eta}\frac{\varepsilon^2 e^{-2(1+\gamma+M)\eta}\beta}{128\gamma}\right) \tag{B.23}$$

$$+ K_0\mathbb{P}\left(B \geq \frac{\varepsilon e^{-(1+\gamma+M)\eta}}{8\eta}\right), \tag{B.24}$$

where the last inequality follows from (B.20), (B.21) and Lemma 19. We can choose $\eta \leq 1$ so that

$$\eta \leq \overline{\eta}_2^U := \frac{\delta\varepsilon^2 e^{-2(1+\gamma+M)}}{384(M^2 C_x^c + (1+\gamma)^2 C_v^c)\mathcal{T}_{\text{rec}}^U}, \tag{B.25}$$

so that the term in (B.22) is less than $\delta/6$, where $C_v^c$, $C_x^c$ are defined in Lemma 10, and then we choose $\beta$ so that

$$\beta \geq \underline{\beta}_2^U := \frac{512d\eta\gamma\log(2^{1/4}e^{1/4}6\delta^{-1}\mathcal{T}_{\text{rec}}^U/\eta)}{\varepsilon^2 e^{-2(1+\gamma+M)\eta}}, \tag{B.26}$$

so that the term in (B.23) is also less than $\delta/6$, and we can choose $\eta$ so that $\eta \leq 1$ and

$$\eta \leq \overline{\eta}_1^U := \frac{\varepsilon e^{-(1+\gamma+M)}}{8B}, \tag{B.27}$$

so that the term in (B.24) is zero.

To complete the proof, let us work on the leading orders of the constants. For the sake of convenience, we hide the dependence on $M$ and $L$ and assume that $M, L = \mathcal{O}(1)$. We also assume that $C_H = \mathcal{O}(1)$. Recall that $0 < \varepsilon \leq \min\{\overline{\varepsilon}_1^U, \overline{\varepsilon}_2^U, \overline{\varepsilon}_3^U\}$, where it is easy to check that It is easy to check that

$$\overline{\varepsilon}_1^U = \sqrt{\frac{C_H + 2 + (m+1)^2}{(C_H + 2)m + (m+1)^2}} r \geq \Omega\left(\frac{C_H^{1/2} r}{C_H^{1/2} m^{1/2} + m + 1}\right) \geq \Omega(r),$$

where we used $m \leq M = \mathcal{O}(1)$ and

$$\overline{\varepsilon}_2^U = 2\sqrt{2}(C_H + 2 + (m+1)^2)^{1/4} \frac{e^{-1/2} r}{m^{1/4}} \geq \Omega\left(\frac{(1 + C_H^{1/4}) r}{m^{1/4}}\right) \geq \Omega\left(\frac{r}{m^{1/4}}\right),$$

and

$$\overline{\varepsilon}_3^U = \frac{\sqrt{m}}{4L\left(\sqrt{C_H + 2} + \frac{m+1}{\sqrt{m}} + \frac{\sqrt{(C_H+2)m}+(m+1)}{8\sqrt{C_H+2+(m+1)^2}}\right)} \geq \Omega\left(\frac{\sqrt{m}}{L\left(1 + \frac{m+1}{\sqrt{m}} + \frac{\sqrt{m}}{m+1}\right)}\right) \geq \Omega(m),$$

where we used the fact that $m + 1 \geq 2\sqrt{m}$. Hence, we can take

$$\varepsilon \leq \min\left\{\mathcal{O}(r), \mathcal{O}\left(\frac{r}{m^{1/4}}\right), \mathcal{O}(m)\right\}.$$

Moreover, $m \leq M = \mathcal{O}(1)$. Hence, we can take

$$\varepsilon \leq \min\{\mathcal{O}(r), \mathcal{O}(m)\}.$$

Next, we recall the recurrence time:

$$\mathcal{T}_{\text{rec}}^U = -\frac{1}{\sqrt{m}} W_{-1}\left(\frac{-\varepsilon^2 \sqrt{m}}{8r^2 \sqrt{C_H + 2 + (m+1)^2}}\right),$$

and since $W_{-1}(-x) \sim \log(1/x)$ for $x \to 0^+$, and we assume $C_H = \mathcal{O}(1)$, we get

$$\mathcal{T}_{\text{rec}}^U = \mathcal{O}\left(\frac{1}{\sqrt{m}} \log\left(\frac{r}{\varepsilon m}\right)\right) \leq \mathcal{O}\left(\frac{|\log(m)|}{\sqrt{m}} \log\left(\frac{r}{\varepsilon}\right)\right).$$

Next, we recall that stepsize $\eta$ satisfies $\eta \leq \min\{1, \overline{\eta}_1^U, \overline{\eta}_2^U, \overline{\eta}_3^U, \overline{\eta}_4^U\}$ and it is easy to check that

$$\overline{\eta}_1^U = \frac{\varepsilon e^{-(1+2\sqrt{m}+M)}}{8B} \geq \Omega\left(\varepsilon e^{-(2m^{1/2}+M)}\right) \geq \Omega(\varepsilon),$$

and

$$\overline{\eta}_2^U = \frac{\delta \varepsilon^2 e^{-2(1+2\sqrt{m}+M)}}{384(M^2 C_x^c + (1 + 2\sqrt{m})^2 C_v^c)\mathcal{T}_{\text{rec}}^U} \geq \Omega\left(\frac{\delta \varepsilon^2 e^{-(4m^{1/2}+2M)}}{(M^2 C_x^c + (1+m)C_v^c)\mathcal{T}_{\text{rec}}^U}\right).$$

Moreover, we have (note that $R = \sqrt{b/m}$ in the definition of $C_x^c, C_v^c$)

$$C_x^c \leq \mathcal{O}\left(\frac{1 + \frac{1}{m} + \frac{d}{\beta}}{m}\right), \qquad C_v^c \leq \mathcal{O}\left(1 + \frac{1}{m} + \frac{d}{\beta}\right),$$

together with $m \leq M = \mathcal{O}(1)$ implies that

$$\overline{\eta}_2^U = \frac{\delta \varepsilon^2 e^{-2(1+2\sqrt{m}+M)}}{384(M^2 C_x^c + (1 + 2\sqrt{m})^2 C_v^c)\mathcal{T}_{\text{rec}}^U} \geq \Omega\left(\frac{m^2 \beta \delta \varepsilon^2}{(md + \beta)\mathcal{T}_{\text{rec}}^U}\right).$$

Moreover,

$$\overline{\eta}_3^U = \frac{2\sqrt{2}m^{1/4}\delta}{3\sqrt{3\beta}M(C_v^d + 2M^2C_x^d + 2B^2 + \frac{4d\sqrt{m}\beta^{-1}}{3})^{1/2}(\mathcal{T}_{\mathrm{esc}}^U)^{1/2}} \geq \Omega\left(\frac{m^{5/4}\delta}{(d+\beta)^{1/2}(\mathcal{T}_{\mathrm{esc}}^U)^{1/2}}\right),$$

where we used $C_x^d \leq \mathcal{O}\left(\frac{d+\beta}{\beta m^2}\right)$ and $C_v^d \leq \mathcal{O}\left(\frac{d+\beta}{\beta m}\right)$, and

$$\overline{\eta}_4^U = \min\left\{1, \frac{2\sqrt{m}}{\hat{K}_2}\frac{d+\overline{A}}{\beta}, \frac{\sqrt{m}\lambda}{\hat{K}_1}\right\} \geq \min\left\{\Omega\left(\frac{m^{1/2}(d+\beta)}{dm^{1/2}+\beta}\right), \Omega(m^{5/2})\right\},$$

where we used $\lambda = \Omega(m)$, $\overline{A} = \Omega(\beta)$, $K_1 = \mathcal{O}(\frac{1}{\beta m})$, $K_2 = \mathcal{O}(1)$, $\hat{K}_1 = \mathcal{O}(\frac{1}{m})$, $\hat{K}_2 = \mathcal{O}(1 + \frac{d}{\beta}\sqrt{m})$, and the minimum between $\frac{m^{1/2}(d+\beta)}{dm^{1/2}+\beta}$ and $m^{5/2}$ is $m^{5/2}$. Hence, we can take

$$\eta \leq \min\left\{\mathcal{O}(\varepsilon), \mathcal{O}\left(\frac{m^2\beta\delta\varepsilon^2}{(md+\beta)\mathcal{T}_{\mathrm{rec}}^U}\right), \mathcal{O}\left(\frac{m^{5/4}\delta}{(d+\beta)^{1/2}(\mathcal{T}_{\mathrm{esc}}^U)^{1/2}}\right), \mathcal{O}(m^{5/2})\right\}.$$

Finally, $\beta$ satisfies $\beta \geq \max\{\underline{\beta}_1^U, \underline{\beta}_2^U\}$, and We have

$$\underline{\beta}_1^U = \frac{256(2C_H m + 4m + (m+1)^2)}{m\varepsilon^2}\left(d\log(2) + \log\left(\frac{6(4m+M^2+1)^{1/2}\mathcal{T}+3}{\delta}\right)\right)$$

$$\leq \mathcal{O}\left(\frac{d+\log((\mathcal{T}+1)/\delta)}{m\varepsilon^2}\right),$$

and

$$\underline{\beta}_2^U = \frac{1024d\eta\sqrt{m}\log(2^{1/4}e^{1/4}6\delta^{-1}\mathcal{T}_{\mathrm{rec}}^U/\eta)}{\varepsilon^2 e^{-2(1+2\sqrt{m}+M)\eta}} \leq \mathcal{O}\left(\frac{d\eta m^{1/2}\log(\delta^{-1}\mathcal{T}_{\mathrm{rec}}^U/\eta)}{\varepsilon^2}\right),$$

where we used $e^{2(1+2\sqrt{m}+M)\eta} = e^{\mathcal{O}(\varepsilon)} = \mathcal{O}(1)$.

Hence, we can take

$$\beta \geq \max\left\{\Omega\left(\frac{d+\log((\mathcal{T}+1)/\delta)}{m\varepsilon^2}\right), \Omega\left(\frac{d\eta m^{1/2}\log(\delta^{-1}\mathcal{T}_{\mathrm{rec}}^U/\eta)}{\varepsilon^2}\right)\right\}.$$

The proof is now complete.

### B.2.2    Proof of Proposition 7

In this section, we focus on the proof of Proposition 7. We adopt some ideas from [BG03, TLR18].
We recall $x_*$ is a local minimum of $F$ and $H$ is the Hessian matrix: $H = \nabla^2 F(x_*)$, and we write

$$X(t) = Y(t) + x_*.$$

Thus, we have the decomposition

$$\nabla F(X(t)) = HY(t) - \rho(Y(t)),$$

where $\|\rho(Y(t))\| \leq \frac{1}{2}L\|Y(t)\|^2$ since the Hessian of $F$ is $L$-Lipschitz (Lemma 1.2.4. [Nes13]).
Then, we have

$$dV(t) = -\gamma V(t)dt - (H(Y(t)) - \rho(Y(t)))dt + \sqrt{2\gamma\beta^{-1}}dB_t,$$
$$dY(t) = V(t)dt.$$

We can write it in terms of matrix form as:

$$d\begin{bmatrix} V(t) \\ Y(t) \end{bmatrix} = \begin{bmatrix} -\gamma I & -H \\ I & 0 \end{bmatrix}\begin{bmatrix} V(t) \\ Y(t) \end{bmatrix}dt + \sqrt{2\gamma\beta^{-1}}\begin{bmatrix} I & 0 \\ 0 & 0 \end{bmatrix}dB_t^{(2)} + \begin{bmatrix} \rho(V(t)) \\ 0 \end{bmatrix}dt,$$

where $B_t^{(2)}$ is a $2d$-dimensional standard Brownian motion. Therefore, we have

$$\begin{bmatrix} V(t) \\ Y(t) \end{bmatrix} = e^{-tH_\gamma}\begin{bmatrix} V(0) \\ Y(0) \end{bmatrix} + \sqrt{2\gamma\beta^{-1}}\int_0^t e^{(s-t)H_\gamma}I^{(2)}dB_s^{(2)} + \int_0^t e^{(s-t)H_\gamma}\begin{bmatrix} \rho(V(s)) \\ 0 \end{bmatrix}ds,$$

557 where

$$H_\gamma = \begin{bmatrix} \gamma I & H \\ -I & 0 \end{bmatrix}, \qquad I^{(2)} = \begin{bmatrix} I & 0 \\ 0 & 0 \end{bmatrix}. \tag{B.28}$$

558 Given $0 \le t_0 \le t_1$, we define the matrix flow

$$Q_{t_0}(t) := e^{(t_0 - t)H_\gamma} \tag{B.29}$$

559 and we also define

$$Z(t) := e^{(t - t_0)H_\gamma} \begin{bmatrix} V(t) \\ Y(t) \end{bmatrix} = Z_t^0 + Z_t^1,$$

560 where

$$Z_t^0 = e^{-t_0 H_\gamma} \begin{bmatrix} V(0) \\ Y(0) \end{bmatrix} + \sqrt{2\gamma\beta^{-1}} \int_0^t e^{(s - t_0)H_\gamma} I^{(2)} dB_s^{(2)}, \tag{B.30}$$

$$Z_t^1 = \int_0^t e^{(s - t_0)H_\gamma} \begin{bmatrix} \rho(V(s)) \\ 0 \end{bmatrix} ds. \tag{B.31}$$

561 Note that

$$Q_{t_0}(t_1)Z_t^0 = e^{-t_1 H_\gamma} \begin{bmatrix} V(0) \\ Y(0) \end{bmatrix} + \sqrt{2\gamma\beta^{-1}} \int_0^t e^{(s - t_1)H_\gamma} I^{(2)} dB_s^{(2)}$$

562 is a martingale. Before we proceed to the proof of Proposition 7, we state the following lemma,
563 which will be used in the proof of Proposition 7.

564 **Lemma 8.** *Assume $\gamma = 2\sqrt{m}$. Define:*

$$\mu_t := e^{-tH_\gamma}(V(0), Y(0))^T, \tag{B.32}$$

$$\Sigma_t := 2\gamma\beta^{-1} \int_0^t e^{(s - t)H_\gamma} I^{(2)} e^{(s - t)H_\gamma^T} ds. \tag{B.33}$$

565 *For any $\theta \in \left(0, \frac{2m\sqrt{m}}{\gamma(2C_H m + 4m + (m+1)^2)}\right)$, and $h > 0$ and any $(V(0), Y(0))$,*

$$\mathbb{P}\left( \sup_{t_0 \le t \le t_1} \|Q_{t_0}(t_1)Z_t^0\| \ge h \right)$$

$$\le \left( 1 - \theta\frac{\gamma(2C_H m + 4m + (m+1)^2)}{2m\sqrt{m}} \right)^{-d} e^{-\frac{\beta\theta}{2}[h^2 - \langle \mu_{t_1}, (I - \beta\theta\Sigma_{t_1})^{-1}\mu_{t_1}\rangle]}.$$

566 Finally, let us complete the proof of Proposition 7.

567 *Proof of Proposition 7.* Since $\|Y(0)\| = \|X(0) - x_*\| \le r$, we know that $\tau > 0$. Fix some
568 $\mathcal{T}_{\text{rec}}^U \le t_0 \le t_1$, such that $t_1 - t_0 \le \frac{1}{2\|H_\gamma\|}$. Then, for every $t \in [t_0, t_1]$,

$$\|Y(t)\| \le \left\| e^{(t_1 - t)H_\gamma} Q_{t_0}(t_1)Z_t \right\| \le e^{\frac{1}{2}} \|Q_{t_0}(t_1)Z_t\|.$$

569 It follows that (with $e^{-1/2} \ge 1/2$)

$$\mathbb{P}(\tau \in [t_0, t_1])$$

$$= \mathbb{P}\left( \sup_{t_0 \le t \le t_1 \wedge \tau} \frac{\|Y(t)\|}{\varepsilon + re^{-\sqrt{m}t}} \ge 1, \tau \ge t_0 \right)$$

$$\le \mathbb{P}\left( \sup_{t_0 \le t \le t_1 \wedge \tau} \frac{\|Q_{t_0}(t_1)Z_t\|}{\varepsilon + re^{-\sqrt{m}t}} \ge \frac{1}{2}, \tau \ge t_0 \right)$$

$$\le \mathbb{P}\left( \sup_{t_0 \le t \le t_1 \wedge \tau} \frac{\|Q_{t_0}(t_1)Z_t^0\|}{\varepsilon + re^{-\sqrt{m}t}} \ge c_0, \tau \ge t_0 \right) + \mathbb{P}\left( \sup_{t_0 \le t \le t_1 \wedge \tau} \frac{\|Q_{t_0}(t_1)Z_t^1\|}{\varepsilon + re^{-\sqrt{m}t}} \ge c_1, \tau \ge t_0 \right),$$
$$\tag{B.34}$$

570 where $c_0 + c_1 = \frac{1}{2}$ and $c_0, c_1 > 0$. We will first bound the second term in (B.34) which will turn out
571 to be zero, and then use Lemma 8 to bound the first term in (B.34).

First, notice that $Z_t^1 \equiv 0$ in the quadratic case and the second term in (B.34) is automatically zero. In the more general case, we will show that the second term in (B.34) is also zero. On the event $\tau \in [t_0, t_1]$, for any $0 \le s \le t_1 \wedge \tau$, we have

$$\|\rho(Y(s))\| \le \frac{L}{2}\|Y(s)\|^2 \le \frac{L}{2}\left(\varepsilon + re^{-\sqrt{m}s}\right)^2.$$

Therefore, for any $t \in [t_0, t_1 \wedge \tau]$, by Lemma 2, we get

$$\left\|Q_{t_0}(t_1)Z_t^1\right\|$$
$$\le \int_0^t \left\|e^{(s-t_1)H_\gamma}\right\| \cdot \|\rho(Y(s))\| ds$$
$$\le \frac{L}{2}\int_0^t \sqrt{C_H + 2 + (m+1)^2(t_1-s)^2} e^{(s-t_1)\sqrt{m}}\left(\varepsilon + re^{-\sqrt{m}s}\right)^2 ds$$
$$\le L\int_0^t \left(\sqrt{C_H+2} + (m+1)(t_1-s)\right)e^{(s-t_1)\sqrt{m}}\left(\varepsilon^2 + r^2 e^{-2\sqrt{m}s}\right) ds$$
$$\le L\int_0^{t_1} \left(\sqrt{C_H+2} + (m+1)(t_1-s)\right)e^{(s-t_1)\sqrt{m}}\left(\varepsilon^2 + r^2 e^{-2\sqrt{m}s}\right) ds$$
$$\le \frac{L}{\sqrt{m}}\left(\left(\sqrt{C_H+2} + \frac{m+1}{\sqrt{m}}\right)\varepsilon^2 + \sqrt{C_H+2}\, r^2 e^{-\sqrt{m}t_1}\right)$$
$$\qquad + L(m+1)r^2 \int_0^{t_1}(t_1-s)e^{(s-t_1)\sqrt{m}}e^{-2\sqrt{m}s} ds$$
$$\le \frac{L}{\sqrt{m}}\left(\left(\sqrt{C_H+2} + \frac{m+1}{\sqrt{m}}\right)\varepsilon^2 + \sqrt{C_H+2}\, r^2 e^{-\sqrt{m}t_1} + (m+1)r^2 t_1 e^{-t_1\sqrt{m}}\right)$$
$$\le \frac{L}{\sqrt{m}}\left(\left(\sqrt{C_H+2} + \frac{m+1}{\sqrt{m}}\right)\varepsilon^2 + \left(\sqrt{(C_H+2)m} + (m+1)\right)r^2 t_1 e^{-t_1\sqrt{m}}\right)$$
$$\le \frac{L}{\sqrt{m}}\left(\sqrt{C_H+2} + \frac{m+1}{\sqrt{m}} + \frac{\sqrt{(C_H+2)m} + (m+1)}{8\sqrt{C_H+2+(m+1)^2}}\right)\varepsilon^2$$

where we used $t_1 \ge t \ge t_0 \ge \mathcal{T}_{\text{rec}}^U \ge \frac{1}{\sqrt{m}}$, and $t_1 e^{-t_1\sqrt{m}} \le \mathcal{T}_{\text{rec}}^U e^{-\mathcal{T}_{\text{rec}}^U \sqrt{m}}$ and the definition of $\mathcal{T}_{\text{rec}}^U$:

$$\sqrt{C_H + 2 + (m+1)^2}\,\mathcal{T}_{\text{rec}}^U e^{-\sqrt{m}\mathcal{T}_{\text{rec}}^U} = \frac{\varepsilon^2}{8r^2}.$$

Consequently, if we take $c_1 = \frac{L}{\sqrt{m}}\left(\sqrt{C_H+2} + \frac{m+1}{\sqrt{m}} + \frac{\sqrt{(C_H+2)m}+(m+1)}{8\sqrt{C_H+2+(m+1)^2}}\right)\varepsilon$, then,

$$\sup_{t_0 \le t \le t_1 \wedge \tau} \frac{\|Q_{t_0}(t_1)Z_t\|}{\varepsilon + re^{-\sqrt{m}t}} \le \frac{1}{\varepsilon}\sup_{t_0 \le t \le t_1 \wedge \tau}\|Q_{t_0}(t_1)Z_t\| \le c_1,$$

which implies that

$$\mathbb{P}\left(\sup_{t_0 \le t \le t_1 \wedge \tau}\frac{\|Q_{t_0}(t_1)Z_t^1\|}{\varepsilon + re^{-\sqrt{m}t}} \ge c_1, \tau \ge t_0\right) = 0.$$

Moreover, $c_0 = \frac{1}{2} - c_1 = \frac{1}{2} - \frac{L}{\sqrt{m}}\left(\sqrt{C_H+2} + \frac{m+1}{\sqrt{m}} + \frac{\sqrt{(C_H+2)m}+(m+1)}{8\sqrt{C_H+2+(m+1)^2}}\right)\varepsilon > \frac{1}{4}$ since it is

assumed that $\varepsilon < \frac{\sqrt{m}}{4L\left(\sqrt{C_H+2} + \frac{m+1}{\sqrt{m}} + \frac{\sqrt{(C_H+2)m}+(m+1)}{8\sqrt{C_H+2+(m+1)^2}}\right)}$.

581 Second, we will apply Lemma 8 to bound the first term in (B.34). By using $V(0) = 0$ and $\|Y(0)\| \le r$
582 and the definition of $\mu_{t_1}$ and $\Sigma_{t_1}$ in (B.32) and (B.33), we get

$$
\begin{aligned}
&\left\langle \mu_{t_1}, (I - \beta\theta\Sigma_{t_1})^{-1}\mu_{t_1} \right\rangle \\
&= \left\langle e^{-t_1 H_\gamma}(V(0), Y(0))^T, (I - \beta\theta\Sigma_{t_1})^{-1} e^{-t_1 H_\gamma}(V(0), Y(0))^T \right\rangle \\
&\le \left( 1 - \theta\frac{\gamma(2C_H m + 4m + (m+1)^2)}{2m\sqrt{m}} \right)^{-1} \left( C_H + 2 + (m+1)^2 t_1^2 \right) e^{-2\sqrt{m}t_1} r^2 \\
&\le 2\left( (C_H + 2)m + (m+1)^2 \right) t_1^2 e^{-2\sqrt{m}t_1} r^2 \\
&\le \frac{1}{32}\frac{(C_H + 2)m + (m+1)^2}{C_H + 2 + (m+1)^2}\frac{\varepsilon^4}{r^2} \le \frac{1}{32}\varepsilon^2,
\end{aligned}
$$

583 by choosing $\theta = \frac{m\sqrt{m}}{\gamma(2C_H m + 4m + (m+1)^2)}$ and $t_1 \ge \mathcal{T}_{\text{rec}}^U \ge \frac{1}{\sqrt{m}}$, and $t_1 e^{-t_1\sqrt{m}} \le \mathcal{T}_{\text{rec}}^U e^{-\mathcal{T}_{\text{rec}}^U}$,
584 and using the definition $\sqrt{C_H + 2 + (m+1)^2}\mathcal{T}_{\text{rec}}^U e^{-\sqrt{m}\mathcal{T}_{\text{rec}}^U} = \frac{\varepsilon^2}{8r^2}$, and we also used $\varepsilon \le$
585 $\sqrt{\frac{C_H + 2 + (m+1)^2}{(C_H + 2)m + (m+1)^2}}r$.

586 Then with the choice of $h = (\varepsilon + re^{-\sqrt{m}t_1})c_0$ and $\theta = \frac{m\sqrt{m}}{\gamma(2C_H m + 4m + (m+1)^2)}$ in Lemma 8, and using
587 the fact that $h = (\varepsilon + re^{-\sqrt{m}t_1})c_0 \ge \varepsilon c_0$, we get

$$
\begin{aligned}
&\mathbb{P}\left( \sup_{t_0 \le t \le t_1 \wedge \tau} \frac{\|Q_{t_0}(t_1)Z_t^0\|}{\varepsilon + re^{-\sqrt{m}t}} \ge c_0, \tau \ge t_0 \right) \\
&\le \mathbb{P}\left( \sup_{t_0 \le t \le t_1} \|Q_{t_0}(t_1)Z_t^0\| \ge \left( \varepsilon + re^{-\sqrt{m}t_1} \right) c_0 \right) \\
&\le \left( 1 - \theta\frac{\gamma(2C_H m + 4m + (m+1)^2)}{2m\sqrt{m}} \right)^{-\frac{2d}{2}} \cdot \exp\left( -\frac{\beta\theta}{2}\left[ h^2 - \langle \mu_{t_1}, (I - \beta\theta\Sigma_{t_1})^{-1}\mu_{t_1} \rangle \right] \right) \\
&\le 2^d \cdot \exp\left( -\frac{\beta\gamma^{-1}m\sqrt{m}\varepsilon^2}{2(2C_H + 4m + (m+1)^2)}\left( c_0^2 - \frac{1}{32} \right) \right) \\
&\le 2^d \cdot \exp\left( -\frac{\beta\gamma^{-1}m\sqrt{m}\varepsilon^2}{128(2C_H + 4m + (m+1)^2)} \right).
\end{aligned}
$$

588 Thus for any $t_0 \ge \mathcal{T}_{\text{rec}}^U$ and $t_0 \le t_1 \le t_0 + \frac{1}{2\|H_\gamma\|}$,

$$
\mathbb{P}(\tau \in [t_0, t_1]) \le 2^d \cdot \exp\left( -\frac{\beta\gamma^{-1}m\sqrt{m}\varepsilon^2}{128(2C_H m + 4m + (m+1)^2)} \right).
$$

589 Fix any $\mathcal{T} > 0$ and recall the definition of the escape time $\mathcal{T}_{\text{esc}}^U = \mathcal{T} + \mathcal{T}_{\text{rec}}^U$. Partition the interval
590 $[\mathcal{T}_{\text{rec}}^U, \mathcal{T}_{\text{esc}}^U]$ using the points $\mathcal{T}_{\text{rec}}^U = t_0 < t_1 < \cdots < t_{\lceil 2\|H_\gamma\|\mathcal{T}\rceil} = \mathcal{T}_{\text{esc}}^U$ with $t_j = j/(2\|H_\gamma\|)$, then we
591 have

$$
\begin{aligned}
\mathbb{P}\left( \tau \in \left[ \mathcal{T}_{\text{rec}}^U, \mathcal{T}_{\text{esc}}^U \right] \right) &= \sum_{j=0}^{\lceil 2\|H_\gamma\|\mathcal{T}\rceil} \mathbb{P}(\tau \in [t_j, t_{j+1}]) \\
&\le (2\|H_\gamma\|\mathcal{T} + 1) \cdot 2^d \cdot \exp\left( -\frac{\beta\gamma^{-1}m\sqrt{m}\varepsilon^2}{128(2C_H m + 4m + (m+1)^2)} \right) \le \delta,
\end{aligned}
$$

592 provided that

$$
\beta \ge \frac{128(2C_H m + 4m + (m+1)^2)\gamma}{m\sqrt{m}\varepsilon^2}\left( d\log(2) + \log\left( \frac{2\|H_\gamma\|\mathcal{T} + 1}{\delta} \right) \right).
$$

593 Finally, plugging $\gamma = 2\sqrt{m}$ into the above formulas and applying the bound on $\|H_\gamma\|$ from Lemma
594 18, the conclusion follows. $\qquad\square$

### B.2.3 Uniform $L^2$ bounds for underdamped Langevin dynamics

596 In this section, we state the uniform $L^2$ bounds for the continuous time underdamped Langevin
597 dynamics ((1.4) and (1.5)) and the discrete time iterates ((1.7) and (1.8)) in Lemma 10, which is a

modification of Lemma 8 in [GGZ18]. The uniform $L^2$ bound for the discrete dynamics (1.7)-(1.8) is used to derive the relative entropy to compare the laws of the continuous time dynamics and the discrete time dynamics, and the uniform $L^2$ bound for the continuous dynamics (1.4)-(1.5) is used to control the tail of the continuous dynamics in Section B.2.1.

Before we proceed, let us first introduce the following Lyapunov function (from the paper [EGZ19]) which will be used in the proof the uniform $L^2$ boundedness results for both the continuous and discrete underdamped Langevin dynamics. We define the Lyapunov function $\mathcal{V}$ as:

$$\mathcal{V}(x,v) := \beta F(x) + \frac{\beta}{4}\gamma^2 \left( \|x + \gamma^{-1}v\|^2 + \|\gamma^{-1}v\|^2 - \lambda\|x\|^2 \right), \tag{B.35}$$

and $\lambda$ is a positive constant less than $1/4$ according to [EGZ19]. We will first show in the following lemma that we can find explicit constants $\lambda \in (0, \min(1/4, m/(M + \gamma^2/2)))$ and $\overline{A} \in (0,\infty)$ so that the drift condition (B.38) is satisfied. The drift condition is needed in [EGZ19], which is applied to obtain the uniform $L^2$ bounds in [GGZ18] (Lemma 8) that implies the uniform $L^2$ bounds in our current setting (the following Lemma 10).

**Lemma 9.** *Let us define:*

$$\lambda = \frac{1}{2}\min(1/4, m/(M + \gamma^2/2)), \tag{B.36}$$

$$\overline{A} = \frac{\beta}{2}\frac{m}{M + \frac{1}{2}\gamma^2}\left(\frac{B^2}{2M + \gamma^2} + \frac{b}{m}\left(M + \frac{1}{2}\gamma^2\right) + A\right), \tag{B.37}$$

*then the following drift condition holds:*

$$x \cdot \nabla F(x) \geq 2\lambda(F(x) + \gamma^2\|x\|^2/4) - 2\overline{A}/\beta. \tag{B.38}$$

The following lemma provides uniform $L^2$ bounds for the continuous-time underdamped Langevin diffusion process $(X(t), V(t))$ defined in (1.4)-(1.5) and discrete-time underdamped Langevin dynamics $(X_k, V_k)$ defined in (1.7)-(1.8).

**Lemma 10** (Uniform $L^2$ bounds). *Suppose parts $(i)$, $(ii)$, $(iii)$, $(iv)$ of Assumption 1 and the drift condition (B.38) hold. $\gamma > 0$ is arbitrary and $\lambda$, $\overline{A}$ are defined in (B.36) and (B.37).*

$(i)$ *It holds that*

$$\sup_{t\geq 0}\mathbb{E}\|X(t)\|^2 \leq C_x^c := \frac{\left(\frac{\beta M}{2} + \frac{\beta\gamma^2(2-\lambda)}{4}\right)R^2 + \beta BR + \beta A + \frac{3}{4}\beta\|V(0)\|^2 + \frac{d+\overline{A}}{\lambda}}{\frac{1}{8}(1-2\lambda)\beta\gamma^2}, \tag{B.39}$$

$$\sup_{t\geq 0}\mathbb{E}\|V(t)\|^2 \leq C_v^c := \frac{\left(\frac{\beta M}{2} + \frac{\beta\gamma^2(2-\lambda)}{4}\right)R^2 + \beta BR + \beta A + \frac{3}{4}\beta\|V(0)\|^2 + \frac{d+\overline{A}}{\lambda}}{\frac{\beta}{4}(1-2\lambda)}, \tag{B.40}$$

$(ii)$ *For any stepsize $\eta$ satisfying:*

$$0 < \eta \leq \overline{\eta}_4^U := \min\left\{1, \frac{\gamma}{\hat{K}_2}(d/\beta + \overline{A}/\beta), \frac{\gamma\lambda}{2\hat{K}_1}\right\}, \tag{B.41}$$

*where*

$$\hat{K}_1 := K_1 + Q_1\frac{4}{1-2\lambda} + Q_2\frac{8}{(1-2\lambda)\gamma^2}, \tag{B.42}$$

$$\hat{K}_2 := K_2 + Q_3, \tag{B.43}$$

*where*

$$K_1 := \max\left\{\frac{32M^2\left(\frac{1}{2}+\gamma\right)}{(1-2\lambda)\beta\gamma^2}, \frac{8\left(\frac{1}{2}M + \frac{1}{4}\gamma^2 - \frac{1}{4}\gamma^2\lambda + \gamma\right)}{\beta(1-2\lambda)}\right\}, \tag{B.44}$$

$$K_2 := 2B^2\left(\frac{1}{2}+\gamma\right), \tag{B.45}$$

*and*

$$Q_1 := \frac{1}{2}c_0\left((5M + 4 - 2\gamma + (c_0 + \gamma^2)) + (1 + \gamma)\left(\frac{5}{2} + c_0(1 + \gamma)\right) + 2\gamma^2\lambda\right),$$
(B.46)

$$Q_2 := \frac{1}{2}c_0\left[\left((1 + \gamma)\left(c_0(1 + \gamma) + \frac{5}{2}\right) + c_0 + 2 + \lambda\gamma^2 + 2(Mc_0 + M + 1)\right) \cdot 2M^2\right.$$
$$\left. + \left(2M^2 + \gamma^2\lambda + \frac{3}{2}\gamma^2(1 + \gamma)\right)\right],$$
(B.47)

$$Q_3 := c_0\left((1 + \gamma)\left(c_0(1 + \gamma) + \frac{5}{2}\right) + c_0 + 2 + \lambda\gamma^2 + 2(Mc_0 + M + 1)\right)B^2 + c_0B^2$$
$$+ \frac{1}{2}\gamma^3\beta^{-1}c_{22} + \gamma^2\beta^{-1}c_{12} + M\gamma\beta^{-1}c_{22},$$
(B.48)

*where*

$$c_0 := 1 + \gamma^2, \qquad c_{12} := \frac{d}{2}, \qquad c_{22} := \frac{d}{3},$$
(B.49)

*we have*

$$\sup_{j \geq 0} \mathbb{E}\|X_j\|^2 \leq C_x^d := \frac{\left(\frac{\beta M}{2} + \frac{\beta\gamma^2(2-\lambda)}{4}\right)R^2 + \beta BR + \beta A + \frac{3}{4}\beta\|V(0)\|^2 + \frac{4(d+\overline{A})}{\lambda}}{\frac{1}{8}(1 - 2\lambda)\beta\gamma^2},$$
(B.50)

$$\sup_{j \geq 0} \mathbb{E}\|V_j\|^2 \leq C_v^d := \frac{\left(\frac{\beta M}{2} + \frac{\beta\gamma^2(2-\lambda)}{4}\right)R^2 + \beta BR + \beta A + \frac{3}{4}\beta\|V(0)\|^2 + \frac{4(d+\overline{A})}{\lambda}}{\frac{\beta}{4}(1 - 2\lambda)}.$$
(B.51)

### B.2.4 Proofs of auxiliary results

*Proof of Lemma 8.* Note that $Q_{t_0}(t_1)Z_t^0$ is a $2d$-dimensional martingale and by Doob's martingale inequality, for any $h > 0$,

$$\mathbb{P}\left(\sup_{t_0 \leq t \leq t_1} \|Q_{t_0}(t_1)Z_t^0\| \geq h\right) \leq e^{-\beta\theta h^2/2}\mathbb{E}\left[e^{(\beta\theta/2)\|Q_{t_0}(t_1)Z_{t_1}^0\|^2}\right]$$
$$= e^{-\beta\theta h^2/2}\frac{1}{\sqrt{\det(I - \beta\theta\Sigma_{t_1})}}e^{\frac{\beta\theta}{2}\langle\mu_{t_1},(I-\beta\theta\Sigma_{t_1})^{-1}\mu_{t_1}\rangle}, \quad \text{(B.52)}$$

where the last line above uses the fact that $Q_{t_0}(t_1)Z_{t_1}$ is a Gaussian random vector with mean

$$\mu_{t_1} = e^{-t_1 H_\gamma}(V(0), Y(0))^T,$$

and covariance matrix

$$\Sigma_{t_1} = 2\gamma\beta^{-1}\int_0^{t_1}\left(e^{(s-t_1)H_\gamma}I^{(2)}\right)\left(e^{(s-t_1)H_\gamma}I^{(2)}\right)^T ds$$
$$= 2\gamma\beta^{-1}\int_0^{t_1}e^{-sH_\gamma}I^{(2)}e^{-sH_\gamma^T}ds.$$

We next estimate $\det(I - \beta\theta\Sigma_{t_1})$ fron (B.52). Let us recall from Lemma 2 that if $\gamma = 2\sqrt{m}$, then we recall from Lemma 2 that,

$$\left\|e^{-tH_\gamma}\right\| \leq \sqrt{C_H + 2 + (m+1)^2t^2} \cdot e^{-\sqrt{m}t},$$

and thus, we have

$$\|\Sigma_{t_1}\| \leq 2\gamma\beta^{-1}\int_0^{t_1}\left(C_H + 2 + (m+1)^2t^2\right)e^{-2\sqrt{m}t}dt \leq \gamma\beta^{-1}\frac{2C_H m + 4m + (m+1)^2}{2m\sqrt{m}}.$$

Therefore we infer that the eigenvalues of $I - \beta\theta\Sigma$ are bounded below by $1 - \theta\frac{\gamma(2C_H m + 4m + (m+1)^2)}{2m\sqrt{m}}$. The conclusion then follows from (B.52). $\qquad\square$

634 *Proof of Lemma 9.* By Assumption 1 (iii), $x \cdot \nabla F(x) \geq m\|x\|^2 - b$. Thus in order to show the drift
635 condition (B.38), it suffices to show that

$$m\|x\|^2 - b - 2\lambda(F(x) + \gamma^2\|x\|^2/4) \geq -2\overline{A}/\beta. \tag{B.53}$$

636 Given the definition of $\lambda$ in (B.36), by Lemma 20, we get

$$
\begin{aligned}
&m\|x\|^2 - b - 2\lambda(F(x) + \gamma^2\|x\|^2/4) \\
&\geq m\|x\|^2 - b - \frac{m}{M + \frac{1}{2}\gamma^2}(F(x) + \gamma^2\|x\|^2/4) \\
&\geq \frac{mM + \frac{1}{4}m\gamma^2}{M + \frac{1}{2}\gamma^2}\|x\|^2 - b - \frac{m}{M + \frac{1}{2}\gamma^2}\left(\frac{M}{2}\|x\|^2 + B\|x\| + A\right) \\
&= \frac{m}{M + \frac{1}{2}\gamma^2}\left(\frac{1}{2}M\|x\|^2 + \frac{1}{4}\gamma^2\|x\|^2 - B\|x\| - \frac{b}{m}\left(M + \frac{1}{2}\gamma^2\right) - A\right) \\
&\geq \frac{m}{M + \frac{1}{2}\gamma^2}\left(-\frac{B^2}{2M + \gamma^2} - \frac{b}{m}\left(M + \frac{1}{2}\gamma^2\right) - A\right) = -2\overline{A}/\beta,
\end{aligned}
$$

637 by the definition of $\overline{A}$ in (B.37). Hence, (B.53) holds and the proof is complete. $\qquad\square$

638 *Proof of Lemma 10.* According to Lemma EC.1 in [GGZ18],

$$
\begin{aligned}
\sup_{t \geq 0} \mathbb{E}\|X(t)\|^2 &\leq \frac{\int_{\mathbb{R}^{2d}} \mathcal{V}(x,v)d\mu_0(x,v) + \frac{d+\overline{A}}{\lambda}}{\frac{1}{8}(1-2\lambda)\beta\gamma^2}, \\
\sup_{t \geq 0} \mathbb{E}\|V(t)\|^2 &\leq \frac{\int_{\mathbb{R}^{2d}} \mathcal{V}(x,v)d\mu_0(x,v) + \frac{d+\overline{A}}{\lambda}}{\frac{\beta}{4}(1-2\lambda)},
\end{aligned}
$$

639 where $\mathcal{V}$ is the Lyapunov function defined in (B.35) and $\mu_0$ is the initial distribution of $(X(0), V(0))$
640 and in our case, $\mu_0 = \delta_{(X(0),V(0))}$ and $\|X(0)\| \leq R$ and $V(0) \in \mathbb{R}^d$, and for any $0 < \eta \leq$
641 $\min\left\{1, \frac{\gamma}{\hat{K}_2}(d/\beta + \overline{A}/\beta), \frac{\gamma\lambda}{2\hat{K}_1}\right\}$ with $\hat{K}_1$ and $\hat{K}_2$ given in (B.42) and (B.43), [3] and according to
642 Lemma EC.5 in [GGZ18], we also have

$$
\begin{aligned}
\sup_{j \geq 0} \mathbb{E}\|X_j\|^2 &\leq \frac{\int_{\mathbb{R}^{2d}} \mathcal{V}(x,v)\mu_0(dx,dv) + \frac{4(d+\overline{A})}{\lambda}}{\frac{1}{8}(1-2\lambda)\beta\gamma^2}, \\
\sup_{j \geq 0} \mathbb{E}\|V_j\|^2 &\leq \frac{\int_{\mathbb{R}^{2d}} \mathcal{V}(x,v)\mu_0(dx,dv) + \frac{4(d+\overline{A})}{\lambda}}{\frac{\beta}{4}(1-2\lambda)}.
\end{aligned}
$$

643 We recall from (B.35) that $\mathcal{V}(x,v) = \beta F(x) + \frac{\beta}{4}\gamma^2(\|x + \gamma^{-1}v\|^2 + \|\gamma^{-1}v\|^2 - \lambda\|x\|^2)$, and
644 $\|X(0)\| \leq R$ and $V(0) \in \mathbb{R}^d$. By Lemma 20, we get

$$\mathcal{V}(x,v) \leq \frac{\beta M}{2}\|x\|^2 + \beta B\|x\| + \beta A + \frac{\beta}{4}\gamma^2(\|x + \gamma^{-1}v\|^2 + \|\gamma^{-1}v\|^2 - \lambda\|x\|^2),$$

645 so that

$$
\begin{aligned}
&\mathcal{V}(X(0), V(0)) \\
&= \frac{\beta M}{2}\|X(0)\|^2 + \beta B\|X(0)\| + \beta A + \frac{\beta}{4}\gamma^2(2\|X(0)\|^2 + 3\gamma^{-2}\|V(0)\|^2 - \lambda\|X(0)\|^2) \\
&\leq \left(\frac{\beta M}{2} + \frac{\beta\gamma^2(2-\lambda)}{4}\right)R^2 + \beta B R + \beta A + \frac{3}{4}\beta\|V(0)\|^2.
\end{aligned}
$$

646 Hence, the conclusion follows. $\qquad\square$

## B.3 Proof of Theorem 4

The proof of Theorem 4 is similar to the proof of Theorem 3. For brevity, we omit some of the details, and only outline the key steps and the propositions and lemmas used for the proof of Theorem 4.

**Proposition 11.** *Fix any $r > 0$ and $0 < \varepsilon < \min\{\bar{\varepsilon}_1^J, \bar{\varepsilon}_2^J\}$, where*

$$\bar{\varepsilon}_1^J := \frac{m_J(\tilde{\varepsilon})}{4C_J(\tilde{\varepsilon})(1 + \|J\|)L(1 + \frac{1}{64C_J(\tilde{\varepsilon})^2})}, \qquad \bar{\varepsilon}_2^J := 8rC_J(\tilde{\varepsilon}). \tag{B.54}$$

*Consider the stopping time:*

$$\tau := \inf\left\{t \geq 0 : \|X(t) - x_*\| \geq \varepsilon + re^{-m_J(\tilde{\varepsilon})t}\right\}.$$

*For any initial point $X(0) = x$ with $\|x - x_*\| \leq r$, and*

$$\beta \geq \frac{128C_J(\tilde{\varepsilon})^2}{m_J(\tilde{\varepsilon})\varepsilon^2}\left(\frac{d}{2}\log(2) + \log\left(\frac{2(1 + \|J\|)M\mathcal{T} + 1}{\delta}\right)\right),$$

*we have*

$$\mathbb{P}_x\left(\tau \in [\mathcal{T}_{rec}^J, \mathcal{T}_{esc}^J]\right) \leq \delta.$$

### B.3.1 Completing the proof of Theorem 4

We first compare the discrete dynamics (1.10) and the continuous dynamics (1.9). Define:

$$\tilde{X}(t) = X_0 - \int_0^t A_J\left(\nabla F(\tilde{X}(\lfloor s/\eta\rfloor\eta))\right)ds + \sqrt{2\gamma\beta^{-1}}\int_0^t dB_s. \tag{B.55}$$

The process $\tilde{X}$ defined in (B.55) is the continuous-time interpolation of the iterates $\{X_k\}$. In particular, the joint distribution of $\{X_k : k = 1, 2, \ldots, K\}$ is the same as $\{\tilde{X}(t) : t = \eta, 2\eta, \ldots, K\eta\}$ for any positive integer $K$.

By following Lemma 7 in [RRT17] and apply the uniform $L^2$ bounds for $X_k$ in Corollary 17 provided that the stepsize $\eta$ is sufficiently small (we apply the bound $\|A_J\| \leq 1 + \|J\|$ to Corollary 17)

$$\eta \leq \bar{\eta}_4^J := \frac{1}{M(1 + \|J\|)^2}, \tag{B.56}$$

we will obtain an upper bound on the relative entropy $D(\cdot\|\cdot)$ between the law $\tilde{\mathbb{P}}^{K\eta}$ of $(\tilde{X}(t) : t \leq K\eta)$ and the law $\mathbb{P}^{K\eta}$ of $(X(t) : t \leq K\eta)$, and by Pinsker's inequality an upper bound on the total variation $\|\cdot\|_{TV}$ as well. More precisely, we have

$$\left\|\tilde{\mathbb{P}}^{K\eta} - \mathbb{P}^{K\eta}\right\|_{TV}^2 \leq \frac{1}{2}D\left(\tilde{\mathbb{P}}^{K\eta}\middle\|\mathbb{P}^{K\eta}\right) \leq \frac{1}{2}C_1K\eta^2, \tag{B.57}$$

where (we use the bound $\|A_J\| \leq 1 + \|J\|$)

$$C_1 := 6(\beta((1 + \|J\|)^2M^2C_d + B^2) + d)(1 + \|J\|)^2M^2, \tag{B.58}$$

where $C_d$ is defined in (B.72).

Let us now complete the proof of Theorem 4. We need to show that

$$\mathbb{P}\left((X_1, \ldots, X_K) \in \mathcal{A}\right) \leq \delta,$$

where $K = \lfloor\eta^{-1}\mathcal{T}_{esc}^J\rfloor$ and $\mathcal{A} := \mathcal{A}_1 \cap \mathcal{A}_2$:

$$\mathcal{A}_1 := \left\{(x_1, \ldots, x_K) \in (\mathbb{R}^d)^K : \max_{k \leq \eta^{-1}\mathcal{T}_{rec}^J}\frac{\|x_k - x_*\|}{\varepsilon + re^{-m_J(\tilde{\varepsilon})k\eta}} \leq \frac{1}{2}\right\},$$

$$\mathcal{A}_2 := \left\{(x_1, \ldots, x_K) \in (\mathbb{R}^d)^K : \max_{\eta^{-1}\mathcal{T}_{rec}^J \leq k \leq K}\frac{\|x_k - x_*\|}{\varepsilon + re^{-m_J(\tilde{\varepsilon})k\eta}} \geq 1\right\}.$$

Similar to the proof in Section B.2.1 and by (B.57), we get

$$\mathbb{P}((X_1, \ldots, X_K) \in \mathcal{A}) \leq \mathbb{P}((X(\eta), \ldots, X(K\eta)) \in \mathcal{A}) + \frac{\delta}{3}, \tag{B.59}$$

provided that

$$\eta \leq \overline{\eta}_3^J := \frac{2\delta^2}{9C_1 \mathcal{T}_{\text{esc}}^J}. \tag{B.60}$$

It remains to estimate the probability of $\mathbb{P}((X(\eta),\dots,X(K\eta)) \in \mathcal{A}_1 \cap \mathcal{A}_2)$ for the non-reversible Langevin diffusion. Partition the interval $[0, \mathcal{T}_{\text{rec}}^J]$ using the points $0 = t_1 < t_1 < \dots < t_{\lceil \eta^{-1}\mathcal{T}_{\text{rec}}^J \rceil} = \mathcal{T}_{\text{rec}}^J$ with $t_k = k\eta$ for $k = 0, 1, \dots, \lceil \eta^{-1}\mathcal{T}_{\text{rec}}^J \rceil - 1$, and consider the event:

$$\mathcal{B} := \left\{ \max_{0 \leq k \leq \lceil \eta^{-1}\mathcal{T}_{\text{rec}}^J \rceil - 1} \max_{t \in [t_k, t_{k+1}]} \|X(t) - X(t_{k+1})\| \leq \frac{\varepsilon}{2} \right\}.$$

Similar to the proof in Section B.2.1, we get

$$\mathbb{P}((X(\eta), \cdots, X(K\eta)) \in \mathcal{A}) \leq \frac{\delta}{3} + \mathbb{P}(\mathcal{B}^c), \tag{B.61}$$

provided that (by applying Proposition 11):

$$\beta \geq \underline{\beta}_1^J := \frac{128 C_J(\tilde{\varepsilon})^2}{m_J(\tilde{\varepsilon})\varepsilon^2} \left( \frac{d}{2}\log(2) + \log\left( \frac{6(1 + \|J\|)M\mathcal{T} + 3}{\delta} \right) \right). \tag{B.62}$$

To complete the proof, we need to show that $\mathbb{P}(\mathcal{B}^c) \leq \frac{\delta}{3}$ in view of (B.59) and (B.61). For any $t \in [t_k, t_{k+1}]$, where $t_{k+1} - t_k = \eta$, we have

$$\|X(t) - X(t_{k+1})\|$$
$$\leq \int_t^{t_{k+1}} \|A_J \nabla F(X(s))\| ds + \sqrt{2\beta^{-1}}\|B_t - B_{t_{k+1}}\|$$
$$\leq \|A_J\|M \int_t^{t_{k+1}} \|X(s) - X(t_{k+1})\| ds + \eta\|A_J \nabla F(X(t_{k+1}))\| + \sqrt{2\beta^{-1}}\|B_t - B_{t_{k+1}}\|$$
$$\leq \|A_J\|M \int_t^{t_{k+1}} \|X(s) - X(t_{k+1})\| ds$$
$$\qquad + \eta\|A_J\| \cdot (M\|X(t_{k+1})\| + B) + \sqrt{2\beta^{-1}}\|B_t - B_{t_{k+1}}\|.$$

By Gronwall's inequality, we get the key estimate:

$$\sup_{t \in [t_k, t_{k+1}]} \|X(t) - X(t_{k+1})\|$$
$$\leq e^{\eta\|A_J\|M} \left[ \eta\|A_J\| \cdot (M\|X(t_{k+1})\| + B) + \sqrt{2\beta^{-1}} \sup_{t \in [t_k, t_{k+1}]} \|B_t - B_{t_{k+1}}\| \right].$$

Then, by following the same argument as in Section B.2.1 and also apply $\|A_J\| \leq 1 + \|J\|$, we can show that $\mathbb{P}(\mathcal{B}^c) \leq \frac{\delta}{3}$ provided that $\eta \leq 1$ and

$$\eta \leq \overline{\eta}_1^J := \frac{\varepsilon e^{-(1+\|J\|)M}}{8(1 + \|J\|)B}, \tag{B.63}$$

and

$$\eta \leq \overline{\eta}_2^J := \frac{\delta\varepsilon^2 e^{-2(1+\|J\|)M}}{384(1 + \|J\|)^2 M^2 C_c \mathcal{T}_{\text{rec}}^J}, \tag{B.64}$$

where $C_c$ is defined in (B.71) and

$$\beta \geq \underline{\beta}_2^J := \frac{512 d\eta \log(2^{1/4}e^{1/4}6\delta^{-1}\mathcal{T}_{\text{rec}}^J/\eta)}{\varepsilon^2 e^{-2(1+\|J\|)M\eta}}. \tag{B.65}$$

To complete the proof, we need work on the leading orders of the constants. We treat $\|J\|$, $M$, $L$ as constant. The argument is similar to the argument in the proof of Theorem 3 and is thus omitted here. The proof is now complete.

### B.3.2 Proof of Proposition 11

Before we proceed to the proof of Proposition 11, let us first state the following two lemmas that will be used in the proof of Proposition 11.

**Lemma 12.** *For any $\theta \in (0, \frac{\lambda_1^J - \tilde{\varepsilon}}{(C_J(\tilde{\varepsilon}))^2})$, $h > 0$ and $y_0 \in \mathbb{R}^d$,*

$$
\mathbb{P}\left( \sup_{t_0 \le t \le t_1} \left\| Q_{t_0}(t_1) Z_t^0 \right\| \ge h \right) \le \left( 1 - \theta \frac{(C_J(\tilde{\varepsilon}))^2}{\lambda_1^J - \tilde{\varepsilon}} \right)^{-d/2} e^{-\frac{\beta\theta}{2}[h^2 - \langle \mu_{t_1}, (I - \beta\theta\Sigma_{t_1})^{-1}\mu_{t_1} \rangle]},
$$

*where $Q_{t_0}(t_1)$ is defined in (B.67), $Z_t^0$ is defined in (B.68), and*

$$
\mu_t := e^{-tA_J H} y_0, \qquad \Sigma_t := 2\beta^{-1} \int_0^t e^{-s(A_J H)} e^{-s(A_J H)^T} ds. \tag{B.66}
$$

**Lemma 13.** *Given $t_0 \le t \le (t_1 \wedge \tau)$, where $\tau$ is the stopping time defined in Proposition 11, we have*

$$
\left\| Q_{t_0}(t_1) Z_t^1 \right\| \le \frac{C_J(\tilde{\varepsilon}) \|A_J\| L}{2} \int_0^t e^{(s-t_1)m_J(\tilde{\varepsilon})} \left( \varepsilon + r e^{-m_J(\tilde{\varepsilon})s} \right)^2 ds,
$$

*where $Q_{t_0}(t_1)$ is defined in (B.67), and $Z_t^1$ is defined in (B.69).*

*Proof of Proposition 11.* We recall $x_*$ is a local minimum of $F$ and $H$ is the Hessian matrix: $H = \nabla^2 F(x_*)$, and we write

$$
X(t) = Y(t) + x_*.
$$

Thus, we have the decomposition

$$
\nabla F(X(t)) = HY(t) - \rho(Y(t)),
$$

where $\|\rho(Y(t))\| \le \frac{1}{2}L\|Y(t)\|^2$ since the Hessian of $F$ is $L$-Lipschitz (Lemma 1.2.4. [Nes13]). This implies that

$$
dY(t) = -A_J HY(t)dt + A_J \rho(Y(t))dt + \sqrt{2\beta^{-1}}dB_t.
$$

Thus, we get

$$
Y(t) = e^{-tA_J H} Y(0) + \sqrt{2\beta^{-1}} \int_0^t e^{(s-t)A_J H} dB_s + \int_0^t e^{(s-t)A_J H} A_J \rho(Y(s))ds.
$$

Given $0 \le t_0 \le t_1$, we define the matrix flow

$$
Q_{t_0}(t) := e^{(t_0-t)A_J H}, \tag{B.67}
$$

and $Z_t := e^{(t-t_0)A_J H} Y_t$ so that

$$
Z_t = e^{-t_0 A_J H} Y(0) + \sqrt{2\beta^{-1}} \int_0^t e^{(s-t_0)A_J H} dB_s + \int_0^t e^{(s-t_0)A_J H} A_J \rho(Y(s))ds.
$$

We define the decomposition $Z_t = Z_t^0 + Z_t^1$, where

$$
Z_t^0 = e^{-t_0 A_J H} Y(0) + \sqrt{2\beta^{-1}} \int_0^t e^{(s-t_0)A_J H} dB_s, \tag{B.68}
$$

$$
Z_t^1 = \int_0^t e^{(s-t_0)A_J H} A_J \rho(Y(s))ds. \tag{B.69}
$$

It follows that for any $t_0 \le t \le t_1$,

$$
Q_{t_0}(t_1) Z_t^1 = \int_0^t e^{(s-t_1)A_J H} A_J \rho(Y(s))ds,
$$

$$
Q_{t_0}(t_1) Z_t^0 = e^{-t_1 A_J H} Y(0) + \sqrt{2\beta^{-1}} \int_0^t e^{(s-t_1)A_J H} dB_s.
$$

The rest of the proof is similar to the proof of Proposition 7. We apply Lemma 13 to bound the term $Q_{t_0}(t_1) Z_t^1$ and apply Lemma 12 to bound the term $Q_{t_0}(t_1) Z_t^0$. By letting $\gamma = 1$ in Proposition 7 and replacing $d$ by $d/2$ due to Lemma 12, and $\|H_\gamma\|$ by $\|A_J H\|$ and using the bounds $\|A_J\| \le (1 + \|J\|)$ and $\|A_J H\| \le (1 + \|J\|)M$, we obtain the desired result in Proposition 11. $\qquad\square$

### B.3.3 Uniform $L^2$ bounds for NLD

In this section we establish uniform $L^2$ bounds for both the continuous time dynamics (1.9) and discrete time dynamics (1.10). The main idea of the proof is to use Lyapunov functions. Our local analysis result relies on the approximation of the continuous time dynamics (1.9) by the discrete time dynamics (1.10). The uniform $L^2$ bound for the discrete dynamics (1.10) is used to derive the relative entropy to compare the laws of the continuous time dynamics and the discrete time dynamics, and the uniform $L^2$ bound for the continuous dynamics (1.9) is used to control the tail of the continuous dynamics in Section B.3.1. We first recall the continuous-time dynamics from (1.9):

$$dX(t) = -A_J(\nabla F(X(t)))dt + \sqrt{2\beta^{-1}}dB_t, \qquad A_J = I + J,$$

where $J$ is a $d \times d$ anti-symmetric matrix, i.e. $J^T = -J$. The generator of this continuous time process is given by

$$\mathcal{L} = -A_J \nabla F \cdot \nabla + \beta^{-1}\Delta \tag{B.70}$$

**Lemma 14.** *Given $X(0) = x \in \mathbb{R}^d$,*

$$\mathbb{E}[F(X(t))] \leq F(x) + \frac{B}{2} + A + \frac{b(M+B)}{m} + \frac{2M\beta^{-1}d(M+B)}{m^2}.$$

Since $F$ has at most the quadratic growth (due to Lemma 20), we immediately have the following corollary.

**Corollary 15.** *Given $\|X(0)\| \leq R = \sqrt{b/m}$,*

$$\mathbb{E}[\|X(t)\|^2] \leq C_c := \frac{MR^2 + 2BR + B + 4A}{m} + \frac{2b(M+B)}{m^2} + \frac{4M\beta^{-1}d(M+B)}{m^3} + \frac{b}{m}\log 3. \tag{B.71}$$

We next show uniform $L^2$ bounds for the discrete iterates $X_k$, where we recall from (1.10) that the non-reversible Langevin dynamics is given by:

$$X_{k+1} = X_k - \eta A_J(\nabla F(X_k)) + \sqrt{2\eta\beta^{-1}}\xi_k.$$

**Lemma 16.** *Given that $\eta \leq \frac{1}{M\|A_J\|^2}$, we have*

$$\mathbb{E}_x[F(X_k)] \leq F(x) + \frac{B}{2} + A + \frac{4(M+B)M\beta^{-1}d}{m^2} + \frac{(M+B)b}{m}.$$

Since $F$ has at most the quadratic growth (due to Lemma 20), we immediately have the following corollary.

**Corollary 17.** *Given that $\eta \leq \frac{1}{M\|A_J\|^2}$ and $\|X(0)\| \leq R = \sqrt{b/m}$, we have*

$$\mathbb{E}[\|X_k\|^2] \leq C_d := \frac{MR^2 + 2BR + B + 4A}{m} + \frac{8(M+B)M\beta^{-1}d}{m^3} + \frac{2(M+B)b}{m^2} + \frac{b}{m}\log 3. \tag{B.72}$$

### B.3.4 Proofs of auxiliary results

*Proof of Lemma 12.* By following the proof of Lemma 8. We get

$$\mathbb{P}\left(\sup_{t_0 \leq t \leq t_1} \|Q_{t_0}(t_1)Z_t^0\| \geq h\right) \leq \frac{1}{\sqrt{\det(I - \beta\theta\Sigma_{t_1})}}e^{-\frac{\beta\theta}{2}[h^2 - \langle \mu_{t_1}, (I-\beta\theta\Sigma_{t_1})^{-1}\mu_{t_1}\rangle]},$$

Recall from (2.3) that for any $\tilde{\varepsilon} > 0$, there exists some $C_J(\tilde{\varepsilon})$ such that for every $t \geq 0$,

$$\left\|e^{-tA_J H}\right\| \leq C_J(\tilde{\varepsilon})e^{-(\lambda_1^J - \tilde{\varepsilon})t},$$

Hence, by the definition of $\Sigma_t$ from (B.66), we get

$$\|\Sigma_t\| \leq 2\beta^{-1}\int_0^\infty (C_J(\tilde{\varepsilon}))^2 e^{-2(\lambda_1^J - \tilde{\varepsilon})t}dt = \frac{\beta^{-1}(C_J(\tilde{\varepsilon}))^2}{\lambda_1^J - \tilde{\varepsilon}}.$$

The rest of the proof follows similarly as in the proof of Lemma 8. $\qquad\square$

*Proof of Lemma 13.* Note that

$$\left\|Q_{t_0}(t_1)Z_t^1\right\| \leq \int_0^t \left\|e^{(s-t_1)A_J H}\right\| \|A_J\| \|\rho(Y(s))\| \, ds,$$

and by applying $\|\rho(Y(t))\| \leq \frac{1}{2}L\|Y(t)\|^2$ and (2.3), and $t_0 \leq t \leq (t_1 \wedge \tau)$ and the definition of the stopping time $\tau$ in Proposition 11, we get the desired result. $\qquad\square$

*Proof of Lemma 14.* Note that if we can show that $F(x)$ is a Lyapunov function for $X(t)$:
$$\mathcal{L}F(x) \leq -\epsilon_1 F(x) + b_1, \tag{B.73}$$
for some $\epsilon_1, b_1 > 0$, then
$$\mathbb{E}[F(X(t))] \leq F(x) + \frac{b_1}{\epsilon_1}.$$

Let us first prove this. Applying Ito formula to $e^{\epsilon_1 t}F(X(t))$, we obtain from Dynkin formula and the drift condition (B.73) that for $t_K := \min\{t, \tau_K\}$ with $\tau_K$ be the exit time of $X(t)$ from a ball centered at 0 with radius $K$ with $X(0) = x$,

$$\mathbb{E}[e^{\epsilon_1 t_K}F(X(t_K))] \leq F(x) + \mathbb{E}\left[\int_0^{t_K} b_1 e^{\epsilon_1 s} ds\right] \leq F(x) + \int_0^t b_1 e^{\epsilon_1 s} ds \leq F(x) + \frac{b_1}{\epsilon_1} \cdot e^{\epsilon_1 t}.$$

Let $K \to \infty$, then we can infer from Fatou's lemma that for any $t$:

$$\mathbb{E}\left[e^{\epsilon_1 t}F(X(t))\right] \leq F(x) + \frac{b_1}{\epsilon_1} \cdot e^{\epsilon_1 t}.$$

Hence, we have

$$\mathbb{E}[F(X(t))] \leq F(x) + \frac{b_1}{\epsilon_1}.$$

Next, let us prove (B.73). By the definition of $\mathcal{L}$ in (B.70), we can compute that
$$\mathcal{L}F(x) = -A_J \nabla F(x) \cdot \nabla F(x) + \beta^{-1}\Delta F(x)$$
$$= -\|\nabla F(x)\|^2 + \beta^{-1}\Delta F(x),$$
since $J$ is anti-symmetric so that $\langle \nabla F(x), J\nabla F(x)\rangle = 0$. Moreover,
$$\|x\| \cdot \|\nabla F(x)\| \geq \langle x, \nabla F(x)\rangle \geq m\|x\|^2 - b, \tag{B.74}$$
implies that
$$\|\nabla F(x)\| \geq m\|x\| - \frac{b}{\|x\|} \geq \frac{1}{2}m\|x\|, \tag{B.75}$$
provided that $\|x\| \geq \sqrt{2b/m}$, and thus
$$\mathcal{L}F(x) \leq -\frac{m^2}{4}\|x\|^2 + \beta^{-1}\Delta F(x) \leq -\frac{m^2}{4}\|x\|^2 + \frac{mb}{2} + \beta^{-1}\Delta F(x), \tag{B.76}$$
for any $\|x\| \geq \sqrt{2b/m}$. On the other hand, for any $\|x\| \leq \sqrt{2b/m}$, we have
$$\mathcal{L}F(x) \leq \beta^{-1}\Delta F(x) \leq -\frac{m^2}{4}\|x\|^2 + \frac{mb}{2} + \beta^{-1}\Delta F(x). \tag{B.77}$$
Hence, for any $x \in \mathbb{R}^d$,
$$\mathcal{L}F(x) \leq -\frac{m^2}{4}\|x\|^2 + \frac{mb}{2} + \beta^{-1}\Delta F(x). \tag{B.78}$$
Next, recall that $F$ is $M$-smooth, and thus
$$\Delta F(x) \leq Md.$$
Finally, by Lemma 20,
$$F(x) \leq \frac{M}{2}\|x\|^2 + B\|x\| + A \leq \frac{M+B}{2}\|x\|^2 + \frac{B}{2} + A.$$
Therefore, we have
$$\mathcal{L}F(x) \leq -\frac{m^2}{2(M+B)}F(x) + \frac{m^2(\frac{B}{2}+A)}{2(M+B)} + \frac{mb}{2} + M\beta^{-1}d.$$
Hence, the proof is complete. $\qquad\square$

*Proof of Corollary 15.* Recall from Lemma 20 that

$$F(x) \geq \frac{m}{2}\|x\|^2 - \frac{b}{2}\log 3,$$

which implies that

$$\|x\|^2 \leq \frac{2}{m}F(x) + \frac{b}{m}\log 3.$$

It then follows from Lemma 14 that

$$\mathbb{E}[\|X(t)\|^2] \leq \frac{2}{m}F(x) + \frac{B}{m} + \frac{2A}{m} + \frac{2b(M+B)}{m^2} + \frac{4M\beta^{-1}d(M+B)}{m^3} + \frac{b}{m}\log 3.$$

Recall that $\|X(0)\| = \|x\| \leq R$ and by Lemma 20 we get $F(x) \leq \frac{M}{2}\|x\|^2 + B\|x\| + A$, and thus

$$\mathbb{E}[\|X(t)\|^2] \leq C_c = \frac{MR^2 + 2BR + B + 4A}{m} + \frac{2b(M+B)}{m^2} + \frac{4M\beta^{-1}d(M+B)}{m^3} + \frac{b}{m}\log 3.$$

$\square$

*Proof of Lemma 16.* Suppose we have

$$\frac{\mathbb{E}_x[F(X_1)] - F(x)}{\eta} \leq -\epsilon_2 F(x) + b_2, \tag{B.79}$$

uniformly for small $\eta$, where $\epsilon_2, b_2$ are positive constants that are independent of $\eta$, then we will first show below that

$$\mathbb{E}_x[F(X_k)] \leq F(x) + \frac{b_2}{\epsilon_2}.$$

We will use the discrete Dynkin's formula (see, e.g. Section 4.2 in [MT92]). Let $\mathbb{F}_i$ denote the filtration generated by $X_0, \ldots, X_i$. Note $\{X_k : k \geq 0\}$ is a time-homogeneous Markov process, so the drift condition (B.79) implies that

$$\mathbb{E}[F(X_i)|\mathbb{F}_{i-1}] \leq (1 - \eta\epsilon_2)F(X_{i-1}) + b_2.$$

Then by letting $r = 1/(1 - \eta\epsilon_2)$, we obtain

$$\mathbb{E}\left[rF(X_i)|\mathbb{F}_{i-1}\right] \leq F(X_{i-1}) + rb_2.$$

Then we can compute that

$$\mathbb{E}\left[r^i F(X_i)|\mathbb{F}_{i-1}\right] - r^{i-1}F(X_{i-1}) = r^{i-1} \cdot \left[\mathbb{E}[rF(X_i)|\mathbb{F}_{i-1}] - F(X_{i-1})\right] \leq r^i b_2. \tag{B.80}$$

Define the stopping time $\tau_{k,K} = \min\{k, \inf\{i : |X_i| \geq K\}\}$, where $K$ is a positive integer, so that $X_i$ is essentially bounded for $i \leq \tau_{k,K}$. Applying the discrete Dynkin's formula (see, e.g. Section 4.2 in [MT92]), we have

$$\mathbb{E}_x\left[r^{\tau_{k,K}}F(X_{\tau_{k,K}})\right] = \mathbb{E}_x\left[F(X_0)\right] + \mathbb{E}\left[\sum_{i=1}^{\tau_{k,K}}\left(\mathbb{E}[r^i F(X_i)|\mathbb{F}_{i-1}] - r^{i-1}F(X_{i-1})\right)\right].$$

Then it follows from (B.80) that

$$\mathbb{E}_x\left[r^{\tau_{k,K}}F(X_{\tau_{k,K}})\right] \leq F(x) + b_2\eta\sum_{i=1}^{k}r^i.$$

As $\tau_{k,K} \to k$ almost surely as $K \to \infty$, we infer from Fatou's Lemma that

$$\mathbb{E}_x\left[r^k F(X_k)\right] \leq F(x) + b_2\eta\sum_{i=1}^{k}r^i,$$

which implies that for all $k$,

$$\mathbb{E}_x\left[F(X_k)\right] \leq F(x) + \frac{b_2\eta}{r-1} = F(x) + \frac{b_2(1 - \eta_2\epsilon_2)}{\epsilon_2} \leq F(x) + \frac{b_2}{\epsilon_2},$$

771    as $r = 1/(1 - \eta_2\epsilon_2)$. Hence we have

$$\mathbb{E}_x\left[F(X_k)\right] \le F(x) + \frac{b_2}{\epsilon_2}.$$

772    It remains to prove (B.79). Note that as $\nabla F$ is Lipschitz continuous with constant $M$ so that:

$$F(y) \le F(x) + \nabla F(x)(y - x) + \frac{M}{2}\|y - x\|^2.$$

773    Therefore,

$$\begin{aligned}
\frac{\mathbb{E}_x[F(X_1)] - F(x)}{\eta} &= \frac{1}{\eta}\left(\mathbb{E}_x\left[F(x - \eta A_J(\nabla F(x)) + \sqrt{2\eta\beta^{-1}}\xi_0)\right] - F(x)\right) \\
&\le -\nabla F(x)A_J\nabla F(x) + \frac{M}{2\eta}\mathbb{E}_x\left[\left\|-\eta A_J(\nabla F(x)) + \sqrt{2\eta\beta^{-1}}\xi_0\right\|^2\right] \\
&= -\|\nabla F(x)\|^2 + \frac{M}{2}\eta\|A_J\nabla F(x)\|^2 + M\beta^{-1}d \\
&\le -\frac{1}{2}\|\nabla F(x)\|^2 + M\beta^{-1}d\,,
\end{aligned}$$

774    provided that $\frac{M}{2}\|A_J\|^2\eta \le \frac{1}{2}$. Similar to the arguments in (B.74)-(B.78), we get

$$\frac{\mathbb{E}_x[F(X_1)] - F(x)}{\eta} \le -\frac{m^2}{8}\|x\|^2 + M\beta^{-1}d + \frac{mb}{4}.$$

775    Finally, by Lemma 20,

$$F(x) \le \frac{M}{2}\|x\|^2 + B\|x\| + A \le \frac{M + B}{2}\|x\|^2 + \frac{B}{2} + A.$$

776    Therefore, we have

$$\frac{\mathbb{E}_x[F(X_1)] - F(x)}{\eta} \le -\frac{m^2}{4(M + B)}F(x) + \frac{m^2(\frac{B}{2} + A)}{4(M + B)} + M\beta^{-1}d + \frac{mb}{4}.$$

777    Hence, the proof is complete.       □

778    *Proof of Corollary 17.* The proof is similar to the proof of Corollary 15 and is thus omitted.    □

## 779   C   Proof of Proposition 5 and Proposition 6

780    *Proof of Proposition 5.* Write $u$ as the corresponding eigenvector of $A_J\mathbb{L}^\sigma$ for the eigenvalue $-\mu_J^* <$
781    $0$, so we have

$$A_J\mathbb{L}^\sigma u = -\mu_J^* u. \tag{C.1}$$

782    Then it follows that

$$(-\mu_J^*)u^*\mathbb{L}^\sigma u = u^*\mathbb{L}^\sigma(-\mu_J^* u) = u^*\mathbb{L}^\sigma A_J\mathbb{L}^\sigma u = u^*(\mathbb{L}^\sigma)^T A_J\mathbb{L}^\sigma u = |\mathbb{L}^\sigma u|^2 + u^*(\mathbb{L}^\sigma)^T J\mathbb{L}^\sigma u,$$

783    where $u^*$ denotes the conjugate transpose of $u$, $(\mathbb{L}^\sigma)^T$ denotes the transpose of $\mathbb{L}^\sigma$, and $(\mathbb{L}^\sigma)^T = \mathbb{L}^\sigma$
784    as $\mathbb{L}^\sigma$ is a real symmetric matrix. It is easy to see that $u^*\mathbb{L}^\sigma u$ is a real number as $(u^*\mathbb{L}^\sigma u)^* = u^*\mathbb{L}^\sigma u$.
785    In addition, $u^*(\mathbb{L}^\sigma)^T J\mathbb{L}^\sigma u$ is pure imaginary, since $(u^*(\mathbb{L}^\sigma)^T J\mathbb{L}^\sigma u)^* = u^*(\mathbb{L}^\sigma)^T J^T\mathbb{L}^\sigma u =$
786    $-u^*(\mathbb{L}^\sigma)^T J\mathbb{L}^\sigma u$ by the fact that $J$ is an anti-symmetric real matrix. Hence, we deduce that

$$u^*(\mathbb{L}^\sigma)^T J\mathbb{L}^\sigma u = 0,$$

787    and it implies that

$$(-\mu_J^*)u^*\mathbb{L}^\sigma u = |\mathbb{L}^\sigma u|^2. \tag{C.2}$$

788    Note $u^*\mathbb{L}^\sigma u \ne 0$ as otherwise $0$ becomes an eigenvalue of $\mathbb{L}^\sigma$ from (C.2), which is a contradiction.
789    In fact, we obtain from (C.2) that $-u^*\mathbb{L}^\sigma u > 0$ as $\mu_J^* > 0$ and $|\mathbb{L}^\sigma u|^2 > 0$.

790    Since $\mathbb{L}^\sigma$ is a real symmetric matrix, we have

$$\mathbb{L}^\sigma = S^T D S, \tag{C.3}$$

for a real orthogonal matrix $S$, where $D = \text{diag}(\mu_1, \mu_2, \ldots, \mu_d)$ with $\mu_1 < 0 < \mu_2 < \ldots < \mu_d$ being the eigenvalues of $\mathbb{L}^\sigma$. Then we obtain

$$\mu_J^* = \frac{|\mathbb{L}^\sigma u|^2}{-u^* \mathbb{L}^\sigma u} = \frac{u^* S^* D^2 S u}{-u^* S^* D S u} = \frac{\sum_{i=1}^d \mu_i^2 |(Su)_i|^2}{\sum_{i=1}^d -\mu_i |(Su)_i|^2}, \tag{C.4}$$

where $(Su)_i$ denotes the $i$-th component of the vector $Su$. Since $\mu_1 < 0 < \mu_2 < \ldots < \mu_d$, we then have $(Su)_1 \neq 0$ as otherwise $-u^* \mathbb{L}^\sigma u = \sum_{i=1}^n -\mu_i |(Su)_i|^2 \leq 0$, which is a contradiction. Therefore, we conclude from (C.4) that

$$\mu_J^* \geq |\mu_1| = \mu^*(\sigma). \tag{C.5}$$

The equality $\mu_J^* = |\mu_1| = \mu^*(\sigma)$ is attained if and only if $(Su)_i = 0$ for $i = 2, \ldots, n$. Or equivalently if and only if the vector $Su = ae_1$ where $a$ is a non-zero constant and $e_1 = [1\ 0\ \ldots\ 0]^T$ is the first basis vector. Since $S^{-1} = S^T$, this is also equivalent to $u = av$ where $v = S^T e_1$ is an eigenvector of $\mathbb{L}^\sigma$ corresponding to the eigenvalue $\mu_1$. Since $u$ and $v$ are related up to a constant, this is the same as saying $v$ is an eigenvector of $A_J \mathbb{L}^\sigma$ satisfying (C.1). Since $v$ is also an eigenvalue of $\mathbb{L}^\sigma$ and $J$ being anti-symmetric, has only purely imaginary eigenvalues except a zero eigenvalue, this is if and only if $Jv = 0$. In other words, the equality $\mu_J^* = |\mu_1| = \mu^*(\sigma)$ is attained if and only if the eigenvector of $\mathbb{L}^\sigma$ corresponding to the negative eigenvalue $\mu_1$ is an eigenvector of $J$ for the eigenvalue 0.

We note finally that Equation (3.5) then readily follows from (3.4) and (C.5). $\qquad\square$

*Proof of Proposition 6.* Write $\tau_{a_1 \to a_2}^{\beta,n}$ for the first time that the continuous-time dynamics $\{X(t)\}$ starting from $a_1$ to exit the region $D_n$. Then by monotone convergence theorem, we have

$$\lim_{R \to \infty} \mathbb{E}\left[\tau_{a_1 \to a_2}^{\beta,n}\right] = \mathbb{E}\left[\tau_{a_1 \to a_2}^{\beta}\right].$$

Hence, for fixed $\epsilon > 0$, one can choose a sufficiently large $n$ such that

$$\left| \mathbb{E}\left[\tau_{a_1 \to a_2}^{\beta,n}\right] - \mathbb{E}\left[\tau_{a_1 \to a_2}^{\beta}\right] \right| < \epsilon. \tag{C.6}$$

We next control the expected difference between the exit times $\hat{\tau}_{a_1 \to a_2}^{\beta,n}$ of the discrete dynamics, and $\tau_{a_1 \to a_2}^{\beta,n}$ of the continuous dynamics, from the bounded domain $D_n$. For fixed $\epsilon$ and large $n$, we can infer from Theorem 4.2 in [GM05] that[4], for sufficiently small stepsize $\eta \leq \bar{\eta}(\epsilon, n, \beta)$,

$$\left| \mathbb{E}\left[\hat{\tau}_{a_1 \to a_2}^{\beta,n}\right] - \mathbb{E}\left[\tau_{a_1 \to a_2}^{\beta,n}\right] \right| < \epsilon. \tag{C.7}$$

Together with (C.6), we obtain for $\eta$ sufficiently small,

$$\left| \mathbb{E}\left[\hat{\tau}_{a_1 \to a_2}^{\beta,n}\right] - \mathbb{E}\left[\tau_{a_1 \to a_2}^{\beta}\right] \right| < 2\epsilon.$$

The proof is therefore complete. $\qquad\square$

# D  Supporting technical lemmas

**Lemma 18.** *Consider the square matrix $H_\gamma$ defined by (2.2). We have*

$$\|H_\gamma\| \leq \sqrt{\gamma^2 + M^2 + 1}.$$

*Proof.* It follows from (B.1) that

$$\|H_\gamma\| = \|T_\gamma\| = \max_i \|T_i(\gamma)\|. \tag{D.1}$$

We also compute

$$\|T_i(\gamma)\|^2 = \lambda_{\max}\left(T_i(\gamma)T_i(\gamma)^T\right) = \lambda_{\max}\left(\begin{bmatrix} \gamma^2 + \lambda_i^2 & -\gamma \\ -\gamma & 1 \end{bmatrix}\right),$$

816    where $\lambda_{\max}$ denotes the largest real part of the eigenvalues. This leads to

$$\|T_i(\gamma)\|^2 = \frac{\gamma^2 + \lambda_i^2 + 1 + \sqrt{(\gamma^2 + \lambda_i^2 + 1)^2 - 4\lambda_i^2}}{2} \le \gamma^2 + \lambda_i^2 + 1.$$

817    Since $m \le \lambda_i \le M$ for every $i$, we obtain

$$\max_i \|T_i(\gamma)\|^2 \le \max_i \left(\gamma^2 + \lambda_i^2 + 1\right) = \gamma^2 + M^2 + 1.$$

818    We conclude from (D.1). $\hfill\square$

819    **Lemma 19.** *Let $B_t$ be a standard $d$-dimensional Brownian motion. For any $u > 0$ and any*
820    *$t_1 > t_0 \ge 0$ with $t_1 - t_0 = \eta > 0$, we have*

$$\mathbb{P}\left(\sup_{t \in [t_0, t_1]} \|B_t - B_{t_1}\| \ge u\right) \le 2^{1/4} e^{1/4} e^{-\frac{u^2}{4d\eta}}.$$

821    *Proof.* Also, by the time reversibility, stationarity of time increments of Brownian motion and Doob's
822    martingale inequality, for any $\theta > 0$ so that $2\theta\eta < 1$, we have

$$\mathbb{P}\left(\sup_{t \in [t_0, t_1]} \|B_t - B_{t_1}\| \ge u\right) = \mathbb{P}\left(\sup_{t \in [0, \eta]} \|B_t - B_0\| \ge u\right)$$

$$\le e^{-\theta u^2} \mathbb{E}\left[e^{\theta \|B_\eta - B_0\|^2}\right]$$

$$= e^{-\theta u^2} (1 - 2\theta\eta)^{-d/2}.$$

823    By choosing $\theta = 1/(4d\eta)$, we get

$$\mathbb{P}\left(\sup_{t \in [t_0, t_1]} \|B_t - B_{t_1}\| \ge u\right) \le \left(1 - \frac{1}{2d}\right)^{-\frac{d}{2}} e^{-\frac{u^2}{4d\eta}}.$$

824    Note that for any $x > 0$, $(1 + \frac{1}{x})^x < e$. Let us define $x > 0$ via

$$1 - \frac{1}{2d} = \frac{1}{1 + x}.$$

825    Then, we get $d = \frac{1+x}{2x}$ and $x = \frac{1}{1 - \frac{1}{2d}} - 1 \le 1$, and

$$\left(1 - \frac{1}{2d}\right)^{-\frac{d}{2}} = \left(\frac{1}{1+x}\right)^{-\frac{1+x}{4x}} = (1+x)^{\frac{1}{4}}(1+x)^{\frac{1}{4x}} \le 2^{1/4} e^{1/4}.$$

826    Hence,

$$\mathbb{P}\left(\sup_{t \in [t_0, t_1]} \|B_t - B_{t_1}\| \ge u\right) \le 2^{1/4} e^{1/4} e^{-\frac{u^2}{4d\eta}}.$$

827    $\hfill\square$

828    **Lemma 20** (See Lemma 2 in [RRT17]). *If parts $(i)$ and $(ii)$ of Assumption 1 hold, then for all*
829    *$x \in \mathbb{R}^d$ and $z \in \mathcal{Z}$,*

$$\|\nabla f(x, z)\| \le M\|x\| + B,$$

830    *and*

$$\frac{m}{3}\|x\|^2 - \frac{b}{2}\log 3 \le f(x, z) \le \frac{M}{2}\|x\|^2 + B\|x\| + A.$$

## Footnotes

[2] The 2-norm of a rank-one matrix $R = uv^*$ should be exactly equal to $\sigma = \|u\|\|v\|$. This follows from the fact that we can write $R = \sigma \tilde{u}\tilde{v}^T$ where $\tilde{u}$ and $\tilde{v}$ have unit norm. This would be a singular value decomposition of $R$, showing that all the singular values are zero except a singular value at $\sigma$. Because the 2-norm is equal to the largest singular value, the 2-norm of $R$ is equal to $\sigma$.

[3] Note that in the definition of $\hat{K}_1, \hat{K}_2$ in [GGZ18], there is a constant $\delta$, which is simply zero, in the context of the current paper.

[4]The Assumption (H2') in Theorem 4.2 of [GM05] can be readily verified in our setting: for both reversible and non-reversible SDE, the drift and diffusion coefficients are clearly Lipschitz; the diffusion matrix is uniformly elliptic; and the domain $D_n$ is bounded and it satisfies the exterior cone condition.