[Reviews · NeurIPS 2020]

Review 1

Summary and Contributions: This paper extends the metastability analysis of Tzen et al 2018 to underdamped Langevin dynamics (ULD) and that with a skew-symmetric streaming matrix (NLD). Authors are able show that the recurrence times of these algorithms improve upon that of vanilla Langevin algorithm. The improvement is of order |log(m)|/sqrt(m) for ULD, where m is the smallest eigenvalue of the Hessian around the local minimum. This improvement is significant. They also show that ULD and NLD has better exit times from the basin of attraction of a local minimum in a simplified setting with two local minima.

Strengths: - Paper is very well written and easy to follow. - Results are new and significant improvements over Tzen et al 2018. More specifically, denoting the smallest eigenvalue of the Hessian of the potential around a local minimum, authors show that ULD achieves a recurrence time T = O(|log(m)|/sqrt(m)) (ignoring epsilon and r). Compared to O(1/m) recurrence time of Tzen et al, this is a significant improvement when m is small. - An exit time analysis from the basin of attraction of a local min is provided for ULD and NLD when the problem is simplified to a double-well potential. Under some conditions, ULD and NLD enjoy better exit times. These results mostly build on existing analysis along similar directions.

Weaknesses: - Paper ends abruptly without a conclusion. - I found the name NLD unfortunate since ULD is also non-reversible. - What happens when condition number is close to 1 or m >= 1 for ULD and NLD? It appears that in this case, ULD performs worse than LD. No discussion is provided for this case. Does LD get better recurrence time compared to ULD in this case? - It is not clear from the presentation if ULD provides any other improvements over LD. Is the dependence on smallest eigenvalue the sole improvement provided by ULD? Perhaps a discussion on which parameters change from LD to ULD would make the analysis more concrete. For example, is there any parameter that gets a worse dependence going from LD to ULD? - Line 220, authors should clarify when does the improvement occur. - How does the exit times of ULD and NLD compare to each other? It would be great if the authors also provide a comparison among these algorithms.

Correctness: Analysis seems to be correct.

Clarity: First 4.5 pages are well written. There are a few issues in the later pages, where the presentation can be further improved. But overall quality is good.

Relation to Prior Work: Relation to prior work is discussed in detail. The closest work to the current paper is Tzen et al 2018 and authors discuss the improvements over the results of that paper in detail.

Reproducibility: Yes

Additional Feedback: Minor comments: 0 Adding a conclusion would be nice for the reader. 1 Authors probably mean isotropic in page 36 2 What is H_gamma in line 114. Used without definition 3 Although H is defined in line 142, authors should redefine it at line 164. 4 M is used in line 165 first, but defined later in line 229 5 In line 172, missing 'which' 6 show'n' missing n in line 198 8 follow'ing' in line 201 =============== I thank the authors for taking the time to answer my questions. After reading other reviewers' comments, I think paper can be accepted as poster.


Review 2

Summary and Contributions: The paper studies two algorithms based on non-reversible versions of Langevin dynamics: underdamped Langevin dynamics (ULD) and Langevin dynamics with non-symmetric drift (NLD). The main result is a quantitative convergence guarantee to local minima. In particular, it is shown that the time scales for ULD is of scales like 1/\sqrt{m} where m is the minimum eigenvalue of the Hessian at the local minimum. In the case of NLD, the dependence 1/m is replaced by 1/m_J, where m_J>m. However the improvement is difficult to quantify in general.

Strengths: - Convergence to local minima in non-convex optimization is a canonical problem. This paper analyzes a new class of non-reversible LD and shows that they offer an advantage over reversible LD. - The analysis requires a certain degree of technical ingenuity although it is substantially based on earlier work.

Weaknesses: - It is unclear why convergence to a local minimum has any relevance for generalization in ML. Under uniform convergence, all that matters is the value of empirical risk achieved. Two optimization algorithms should be compared for the value they achieve, not for their ability to reach local minima. While is some cases the two properties correlate, there is no general reason they should.

Correctness: The approach and results seem sound.

Clarity: The paper is quite clear.

Relation to Prior Work: Earlier work is reviewed.

Reproducibility: Yes

Additional Feedback:


Review 3

Summary and Contributions: After authors' feedback: The authors' response to my question about the flaw in the proof is satisfactory. I also think that the results show some acceleration effect of underdamped dyanmics (albeit built upon well-known facts), which is worth letting the community know. Therefore, I lean towards acceptance. ############################# The paper studies the non-convex optimization problem using underdamped Langevin diffusion (ULD) and non-reversible variants of the Langevin diffusion (NLD). The main contributions are two-fold. On the one hand, a metastability result is shown for both ULD and NLD, depending on the eigen-structure of the associated non-symmetric matrices. When the momentum parameter is chosen optimally, this leads to an $O(\sqrt{m})$ acceleration. On the other hand, the asymptotic acceleration effects of ULD and NLD are also established for the exit time of local minima, with an improved factor.

Strengths: The acceleration effect of second-order diffusions on stochastic optimization is an important question, and the paper provide some interesting insights via a metastability result and an exit time result. I believe a result of this type could be worth publishing, if the issues about the proofs were resolved and some deeper results were obtained.

Weaknesses: The results established in the paper are relatively straightforward, and being built upon existing techniques. Given the third-order smoothness and the local nature of the result, the metastability theorems are essentially characterizing the behavior of underdamped or asymmetric versions of Ornstein-Uhlenbeck processes. The metastability proof techniques follow from [TLR18], and the discretization error analysis is directly using [GGZ18] (which is indeed problematic here. See comments below). The exit-time results are somewhat more interesting, though they are indeed directly applying existing works on Eyring-Kramers formulae.

Correctness: The results are mostly straightforward to understand, and I am certain that a version of the claimed result shuold be correct, albeit with potentially worse dependency on $\eta$ and other parameters. There is a serious issue of "proof by ghost reference" that the authors need to address: In page 4 of the appendix, the authors said "It is derived in the proof of Lemma EC.6 in [GGZ18] that the relative entropy is bounded by ...". This appears to be a crucial step in the discrete-time analysis. However, this step has several issues, as explained below: - First, the paper [GGZ18] does not have a Lemma EC.6. - From my reading of [GGZ18], the relevant result is Lemma 14 therein. However, when taking into account the dependency on the parameters $(K, \eta)$ the KL divergence upper bound in [GGZ18] has terms of order $O (K^3 \eta^4 + K \eta^2 + \delta K \eta)$ (See page 50 of the paper). I am not sure how the claimed bound of order $O (K \eta^3)$ can be correct given this. I would suggest the authors to write a self-contained proof for this theorem, with explicit dependency on problem parameters.

Clarity: The writing needs improvement in several places. To list a few of them: - Assumption (i), (ii) are about stochastic gradient settings, but the rest part of the paper turns out to be assuming a full gradient oracle. This leads to confusion. - Theorems 3,4 are stated in a complicated way, which makes it difficult to see the acceleration effect easily. - The claims in Section 3 are not formally proven. It would be better to write them into theorems and state the assumptions and the results on discrete-time processes clearly.

Relation to Prior Work: The relation to existing works are mostly discussed. The references to two pioneering works are missing, which are the origins of the discretization proof techniques used in the paper: A. Dalalyan, "Theoretical guarantees for approximate sampling from smooth and log-concave densities" A. Durmus and E. Moulines, "Nonasymptotic convergence analysis for the unadjusted Langevin algorithm"

Reproducibility: Yes

Additional Feedback: The paper initiate an interesting study, but there are several issues, and the results are generally not quite novel. I would encourage the authors to fix the issues and dig deeper into the following directions, to see the acceleration effect of high-order schemes: - Is discretization error really needed in the meta-stability analysis? In my opinion, the martingale-based proof can carry out directly in discrete time, and the entire trajectory will be close to a discrete-time linear random dynamical system (as analogous to the OU process used in the paper). Direct bounding the discretization error may lead to unncessarily strong requirements on the step size. - In the exit time part, what will happen when the potential is not a double-well? Can we see a similar acceleration phenomenon of the second-order dyanmics? This could possibly lead to more interesting and novel results.


Review 4

Summary and Contributions: This is a theoretical paper and contains multiple theorems and lemmas. It studies two processes for nonconvex optimization: the underdamped Langevin dynamics (ULD) and the Langevin dynamics with a non-symmetric drift (NLD). It improves the previous results in TLR18.

Strengths: 1. Theoretical results are very thorough and comprehensive; 2. It has improvement over TLR18;

Weaknesses: 1. In terms of the techniques, what are the novel techniques for proof beyond TLR18? In terms of conclusion, what is the key change to obtain a better rate from Overdamped LD to ULD and NLD? 2. Is the goal to minimize $\overline{F}$ or $F$? It seems we need to minimize $\overline{F}$, but the discussions are all for $F$. 3. The assumptions in Assumption 1 seem very strong and limited. For example, is the loss function being twice continuously differentiable practical? 4. What is the choice of J in practice? 5. Based on theoretical analysis, can you provide some guidelines to use ULD and NLS in practice? Can you demonstrate some empirical studies using some simple examples?

Correctness: The claims and method are correct. There is no empirical study.

Clarity: The paper is very theoretical and conclusion is clear. But it is hard to understand the insights.

Relation to Prior Work: I would like to see more discussions and comparisons with previous developments. It may emphasize what re the difference between overdamped LD and ULD and NLD. This difference can lead to a better performance in practice.

Reproducibility: Yes

Additional Feedback:

[Author Response · NeurIPS 2020]

We thank the reviewers for their careful consideration and their feedback, our replies are provided below. We believe that we addressed all the raised issues, the detailed responses are given below.

**Novelty of our analysis and comparison to [TLR18]**: Our analysis is significantly more complicated compared to the LD case in [TLR18] due to non-reversibility, and requires us to develop new estimates, e.g. Lemma 2, where the eigenvalue and the norm estimates require a significant amount of work because the forward iterations correspond to *non-symmetric* matrices $H_\gamma$ (defined in (2.2)) and achieving the acceleration behavior requires careful estimates. The analysis here also requires us to establish novel uniform $L^2$ bounds for NLD in both continuous and discrete times. We have also new results and insights about the mean exit times for ULD and NLD, which is a difficult problem to study.

**R.1: (1)** We will add a conclusion section to summarize our paper. **(2)** The name NLD is indeed a bit unfortunate but the name "non-reversible" for such dynamics is standard, see e.g. "Duncan, A.B., Lelièvre, T. & Pavliotis, G.A. Variance Reduction Using Nonreversible Langevin Samplers. *J Stat Phys* 163, 457-491 (2016)". **(3)** When condition number is close to 1 or $m$ large, ULD may not improve upon LD. **(4)** We will add discussions on dependence on other parameters in the revision. We focus on the comparison in $m$ since it is a natural choice. For example, for convergence rate to Gibbs distribution, it is known that for $m$-strongly convex objectives the continuous-time overdamped Langevin diffusion has rate $e^{-mt}$ in $\mathcal{W}_2$ independent of

other parameters. **(5)** Line 220. The improvement will occur if $C_J(\tilde{\varepsilon}) = \tilde{O}(1)$. **(6)** We will add discussions comparing ULD and NLD. For example, when smallest eigenvalue $m$ is close to the largest eigenvalue $M$, NLD will not improve much upon LD, but ULD will improve upon LD if $m$ is small and ULD will be faster than NLD. The figure on the right is an example for training fully-connected neural networks on MNIST where ULD was faster when both methods were tuned. **(7)** $H_\gamma$ is defined in (2.2). We will define it earlier than Line 114. We will also define $H$ and $M$ more appropriately. **(8)** We will correct all the typos the reviewer pointed out and add clarifications to all the bullet points.

**R.2:** We thank the reviewer for the insightful comments. We absolutely agree with the reviewer that smaller value of empirical risk achieved, better generalization will be in general. However, for many interesting problems such as deep learning with modern neural networks, it has been empirically found that most local minima are equivalent in the sense that they lead to similar generalization performance and that finding a global minima may sometimes lead to overfitting (see e.g. the paper "The Loss Surfaces of Multilayer Networks" by Choromanska et al.). Therefore, there is also incentive to find a local minima to achieve good generalization performance where our results would be relevant.

**R.3: (2)** We sincerely apologize for mis-citing Lemma EC.6 in [GGZ18]. Regarding Lemma 14 in [GGZ18], we would like to point out that in our current paper, our ULD uses the Cheng et al. discretization of underdamped Langevin diffusion as in [CCBJ17]; therefore it corresponds to Lemma 18 in [GGZ18] because in [GGZ18] two discretizations of underdamped Langevin diffusion are considered: Euler discretization (Lemma 14), and Cheng et al. discretization (Lemma 18). Indeed, the proof of Lemma 18 provides the bound $O(K\eta^3)$ for KL divergence; see Equation (D.20) on p.63 of [GGZ18]). We will provide a self-contained proof in the revision. **(3)** Carrying out metastability analysis without relying on discretization error is a good idea, and is worth exploring in the future. One reason we follow the current approach is to make it easier for us to compare our results with [TLR18] and show advantage and improvement when breaking the reversibility. If the improvement comes from avoiding discretization error, it might confuse the readers and undermine and main message of our paper. **(4)** In the exit-time part, we do not know any rigorous results for continuous-time Langevin beyond the double well example. Analyzing the behavior of these processes around a saddle point becomes very hard as the surface that contain the saddle point is characteristic, i.e. the drift and the normal to the surface is orthogonal which makes standard boundary layer approaches inapplicable (see Sec 5 of [BR16]) and we agree with the reviewer that exploring beyond the double well example will be a very interesting research direction to pursue which would lead to a major breakthrough in this research area. **(5)** We will cite the papers A. Dalalyan, "Theoretical guarantees for approximate sampling from smooth and log-concave densities" A. Durmus and E. Moulines, "Nonasymptotic convergence analysis for the unadjusted Langevin algorithm". Thanks for bringing this up!

**R.4: (2)** Our analysis is for the empirical risk $F$, however it is straightforward to obtain standard generalization bounds to get results for the population risk $\overline{F}$ by an analysis similar to [TLR18]. **(3)** We agree that it would be interesting to relax the twice continuously differentiability assumption, however we note that due to the difficulty of analyzing ULD and NLD algorithms, even for such smooth functions many basic questions are open such as the mean exit time or a sharp characterization of time it takes to be in a neighborhood of a local minima. **(4-5)** In practice, the matrix $J$ can be chosen as a random anti-symmetric matrix. For quadratic objectives, there is a

formula for optimal $J$ matrix (see the paper "Optimal non-reversible linear drift for the convergence to equilibrium of a diffusion" by Lelievre et al, 2013). We can take the parameter $\gamma = 2\sqrt{m}$ as predicted by our theory (Lemma 2) for quadratics. In the figure, we compare ULD and NLD to LD for the double well example with random initialization over 100 runs where $J$ is chosen randomly and $\gamma = 2\sqrt{m}$. In this simple example, we observe NLD and ULD have smaller mean exit times (from a barrier) compared to LD. We will add these discussions in the revised version.

[Meta-Review · NeurIPS 2020]

Three reviewers agree that this submission represents an important contribution to the field. However, after reviewing the rebuttal, several still expressed concerns about technical novelty. Please be sure to carefully review and address the concerns of all reviewers in the revision.